# FEDERATED LEARNING NODES CAN RECONSTRUCT PEERS' IMAGE DATA

## ABSTRACT

Federated learning (FL) is a privacy-preserving machine learning framework that enables multiple nodes to train models on their local data and periodically average weight updates to benefit from other nodes' training. Each node's goal is to collaborate with other nodes to improve the model's performance while keeping its training data private. However, this framework does not guarantee data privacy. Prior work has shown that the gradient-sharing steps in FL can be vulnerable to data reconstruction attacks from an honest-but-curious central server. In this work, we show that an honest-but-curious node/client can also launch attacks to reconstruct peers' image data through gradient inversion, presenting a severe privacy risk. We demonstrate that a single client can silently reconstruct other clients' private images using diluted information available within consecutive updates. We leverage state-of-the-art diffusion models to enhance the perceptual quality and recognizability of the reconstructed images, further demonstrating the risk of information leakage at a semantic level. This highlights the need for more robust privacy-preserving mechanisms that protect against silent client-side attacks during federated training. Our source code and pretrained model weights are available at `https://anonymous.4open.science/r/curiousclient-5B6F`.

## 1 INTRODUCTION

Federated learning (FL) has attracted significant attention as a promising approach to privacy-preserving machine learning (McMahan et al., 2017; Kairouz et al., 2021). In this framework, a central server coordinates training by multiple clients. In each training round, the server broadcasts a shared model to clients. Clients may train for a single local iteration using the Federated Stochastic Gradient Descent (FedSGD) protocol, or for multiple local iterations under the Federated Averaging (FedAvg) protocol. Each client then returns the resulting gradient to the server, which averages all the gradients and updates the model. This approach enables each participant to benefit from a model trained on more data without sharing its own data. FL has the potential to revolutionize collaborative efforts in real-world applications such as healthcare and finance, enabling participants to train better models without compromising data privacy (Li et al., 2020a).

Despite the intent to protect privacy through FL, prior works have shown that an *honest-but-curious server* can reconstruct a client's training data via gradient inversion. This is done by adjusting a dummy input to the model until its resulting gradient closely matches the gradient sent by the client (Zhu et al., 2019; Geiping et al., 2020; Kariyappa et al., 2023). Meanwhile, studies on *malicious clients* have shown that a client can disrupt federated training by sending adversarial data to the server (Blanchard et al., 2017; Shi et al., 2022). However, far less attention has been given to the potential for clients to reconstruct others' data via gradient updates while honestly participating in the federated training. This may be partially attributed to an intuition that peer clients can hardly reconstruct meaningful data while adhering to the FL protocol due to the diluted information buried in the heavily aggregated gradient updates. While Wu et al. (2024) have assumed the *honest-but-curious clients* threat model for a related model inversion task, this inherently more difficult task has not demonstrated substantial privacy concerns in terms of image reconstruction quality as seen in gradient inversion attacks launched by honest-but-curious servers.

Our work explores the extent to which a single client can reconstruct high-quality training data from other clients via gradient inversion while adhering to FL protocols. Our work demonstrates

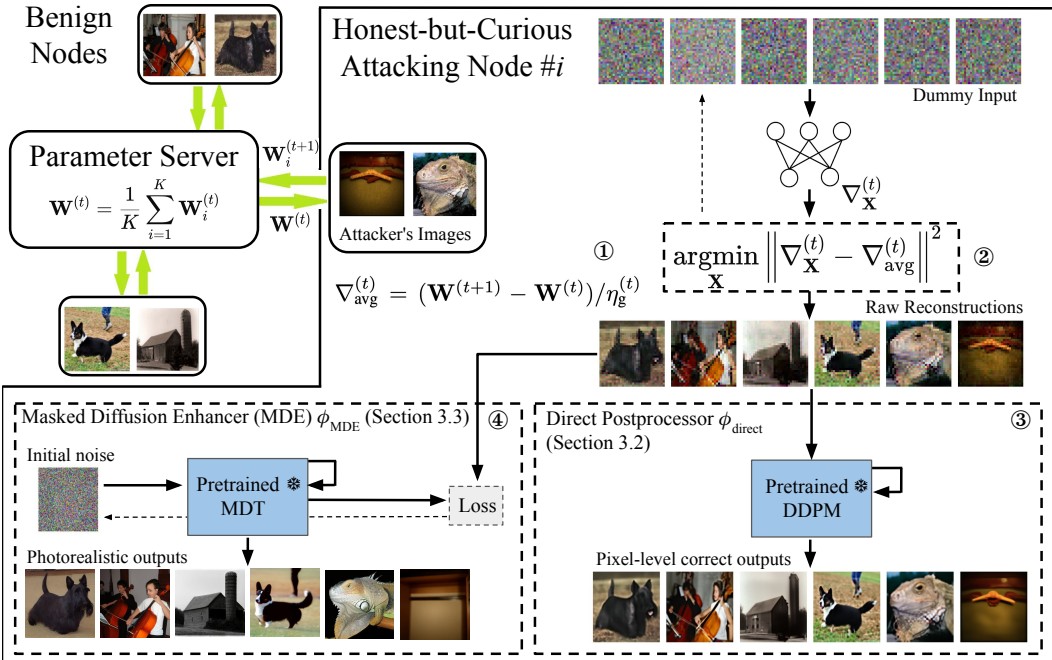

Figure 1: Overview of the proposed honest-but-curious client attack. The attacker participates in two consecutive training rounds to obtain the global model's gradient update by ① differencing the model weights. The attacker then ② inverts the update to obtain each client's training data. To address the challenge of recovering high-quality images from diluted information hidden in the global gradient update, the raw reconstructions are postprocessed using either ③ a direct technique respecting pixel-level correctness or ④ a semantic technique focusing on producing photorealistic images.

that the global model updates—broadcasted by the central server to all clients—contain enough information for a client to reconstruct other clients' image data, regardless of whether model updates are computed under FedSGD or FedAvg. This risk is particularly concerning, as it implies that every participating node, and not just the server, poses a potential privacy threat. Figure 1 depicts the proposed client-to-client attack, where the attacker exploits weight updates between consecutive training rounds to reconstruct training images. By participating in two consecutive training rounds and comparing the global model's weights, a single client extracts the averaged gradient of all participants in the earlier round. Unlike server-side gradient inversion (Zhu et al., 2019; Geiping et al., 2020; Yue et al., 2023), this attack requires isolating individual data from a heterogeneous mixture of gradient updates. Despite this challenge, we show that the attacker can reconstruct images from every other client.

We further propose to utilize two image postprocessing techniques based on diffusion models to improve the quality and recognizability of the attack's raw reconstructed images. Our first technique uses a pretrained masked diffusion transformer (MDT) (Gao et al., 2023) to generate high-quality images that resemble the raw attack results on a semantic level. Our second technique uses denoising diffusion probabilistic models (DDPMs) (Ho et al., 2020) to enhance the raw reconstructions at the pixel level through super resolution and denoising (Kawar et al., 2022). This paper's contributions are fourfold.

1. We experimentally demonstrate that honest-but-curious clients participating in FedAvg (McMahan et al., 2017) can exploit the model update process to reconstruct high-quality image data from peer nodes, highlighting an understudied privacy risk in FL.

2. The proposed masked diffusion enhancer (MDE) generates sharp, high-resolution images from the low-resolution, color-aliased raw reconstructions. MDE's output resembles a target image on a semantic level, preserving its geometric shape and perceptual features with photorealistic quality.

3. The proposed DDPM-based image postprocessing simultaneously denoises and upsamples raw reconstructed images. This improves image resolution and object recognizability, achieving strong pixel-wise similarity to ground-truth images.

4. We theoretically contrasted gradient inversion attacks launched by a client against a hypothetical server or a super client. Their equivalence is stronger with fewer local iterations, more training examples per client, fewer clients, and more training examples across all clients. Our theoretical result justifies the proposed use of an equivalent super-client for the gradient inversion attacks, in which the knowledge of the number of clients and per-client information is not needed.

## 2 RELATED WORK

**Server-Side Gradient Inversion.** The assumption that FL inherently protects data privacy has been challenged by studies exposing vulnerabilities to gradient inversion attacks from the central server (Zhu et al., 2019; Geiping et al., 2020). These attacks exploit the gradients shared by clients to reconstruct private training data. They revealed that by iteratively updating a dummy input to produce a gradient similar to a given target gradient, the server could generate images closely resembling the participant's original training data. Various defense mechanisms have been proposed to protect against these attacks, including gradient compression, perturbation, and differential privacy techniques (Zhang et al., 2020; Sun et al., 2021). Despite these efforts, recent studies have shown that these defenses may not effectively prevent training data from being meaningfully reconstructed. For example, Yue et al. (2023) overcame these defense methods by leveraging latent space reconstruction and incorporating generative models to remove distortion from reconstructed images.

**Client-Side Model Inversion.** While the majority of research has focused on server-side attacks, the potential for client-side attacks has been less examined. FedInverse (Wu et al., 2024) investigated model inversion attacks, where a client exploits the model's overfitting to reconstruct training data. This approach relies on manipulating the model rather than directly reconstructing other clients' data. Similar to our work's takeaway, their results demonstrated that clients can reconstruct peers' images without disrupting the training process. However, due to the challenging nature of model inversion, their method produces reconstructed images far less similar to the target than those from gradient inversion attacks.

**Malicious Client Attacks.** A parallel research direction focuses on attacks where a malicious client interferes with the FL process (Blanchard et al., 2017; Shejwalkar et al., 2022). Specifically, malicious clients can manipulate the model updates by using poisoned data or sending poisoned gradients to the server to impede convergence. Meanwhile, researchers have shown that malicious modifications can compromise privacy easily (Fowl et al., 2021; Wen et al., 2022). While this introduces unique security challenges in FL, our attack does not disrupt the training process and is difficult to be detected by the server or other clients.

## 3 PROPOSED ATTACK BY CURIOUS CLIENTS

In this section, we present the gradient inversion attack which allows an honest-but-curious client to reconstruct image data from other clients. To enhance this reconstruction, two postprocessing methods are introduced to achieve both fine-grained quality and perceptual realism. The first postprocessing method improves the images at the pixel level with enhanced details. The second method, built on a masked diffusion enhancer, refines the images at the semantic level and produces photorealistic reconstructions.

### 3.1 ATTACK FRAMEWORK

**Threat Model.** We consider an honest-but-curious client (or curious client, for simplicity) attacker. It aims to reconstruct other clients' training data while following FL's protocol. The attacker does not disrupt the model training process. Unlike a server-side attacker, the curious client does not have direct access to the gradients from other clients. However, it receives an updated version of the shared model from the server at each communication round. The clients may train for multiple local iterations between global averaging rounds, following the FedAvg protocol. We follow the assumption of Yue et al. (2023) that each client trains for $\tau$ iterations on a single batch of images in

each local iteration/update round and that the class labels have been analytically inverted as in Ma et al. (2023). The attacker may not know the number of clients in each training round but can correctly guess the total number of training images. Additionally, we assume as in Li et al. (2020b); Huang et al. (2020) that all client updates in a given round have been computed using the same learning rate, which is applied locally if each client transmits a model update, as shown in Eq. (2). It may also be applied globally if clients transmit raw gradients to the server, as discussed in Appendix B. We target cross-silo FL scenarios, in which a small number of clients collaborate to overcome data scarcity. For example, a group of hospitals may use FL to develop a classifier for rare diseases from CT scans, where each has limited training examples and images cannot be directly shared due to privacy concerns. We assume that the system is designed to prioritize model accuracy and uses synchronous gradient updates. Clients are not edge devices and have sufficient computational resources to perform the optimization process while participating in FL.

We describe the FL process to be attacked as follows. The $k$th client at time $t$ uses a batch of size $N_k$ to compute its local weights $\mathbf{W}_k^{(t,u)}$ across all local iterations $u$ until $u = \tau$, where $\tau$ is the number of local training iterations. Each client's final local weight can be written as:

$$\mathbf{W}_k^{(t,\tau)} = \mathbf{W}^{(t)} - \frac{\eta_\text{g}}{N_k}\Delta_k^{(t)}, \qquad \Delta_k^{(t)} = \sum_{u=0}^{\tau-1}\sum_{i=1}^{N_k}\nabla\ell(\mathbf{W}_k^{(t,u)};\mathbf{X}_{k,i},\mathbf{Y}_{k,i}), \qquad (1)$$

where $\mathbf{W}^{(t)}$ is the global model parameters at time $t$, $\nabla\ell(\cdot)$ is the gradient of the loss function, the doubly indexed $\mathbf{X}_{k,i} \in \mathbb{R}^{C \times H \times W}$, $\mathbf{Y}_{k,i} \in \mathbb{R}$ are the $i$th training image and label from client $k$, respectively, and $C$, $H$, and $W$ are the number of channels, the height, and the width of the images. The server generates the global weights by a weighted average of all clients' final local weights, namely, $\mathbf{W}^{(t+1)} = \frac{1}{N}\sum_{k=1}^{K}N_k\mathbf{W}_k^{(t,\tau)}$, where $N = \sum_{k=1}^{K}N_k$ is the total number of training examples across $K$ clients, with each client having a fixed minibatch of $N_k$ images. Substituting $\mathbf{W}_k^{(t,\tau)}$ into the expression for $\mathbf{W}^{(t+1)}$, we obtain the global weight update equation:

$$\mathbf{W}^{(t+1)} = \mathbf{W}^{(t)} - \frac{\eta_\text{g}}{N}\sum_{k=1}^{K}\Delta_k^{(t)}. \qquad (2)$$

We note that scaling each client's update by its number of training images $N_k$ causes the gradient of each training image $\mathbf{X}_{k,i}$ to be weighted equally in the global update.

Our approach to reconstructing data from the global model updates builds upon traditional gradient inversion and includes extra initialization and calculation steps to separate individual training images from the averaged global update. Our attacker engages in two consecutive rounds of FedAvg and obtains two versions of the global model parameters, $\mathbf{W}^{(t)}$ and $\mathbf{W}^{(t+1)}$. By computing the difference between successive model weights, the attacker can infer the gradient used for the global model update: $\nabla_\text{avg}^{(t)} = (\mathbf{W}^{(t+1)} - \mathbf{W}^{(t)})/\eta_\text{g}^{(t)}$, where $\eta_\text{g}^{(t)}$ is the globally-determined learning rate for round $t$.

To reconstruct training images, our attacker approximates the global model update as that of a single super-client. In Appendix A, we justify mathematically that this approximation is close when the number of clients $K$, the number of local iterations $\tau$, or the learning rate $\eta_g$ is small, or the number of training images from all clients $N$ is large. In this way, the curious client attack is significantly simplified as it does not need the knowledge of the number of clients and per-client information.

The attacker initializes dummy image data $\mathbf{X} \in \mathbb{R}^{N \times C \times H \times W}$ and labels $\mathbf{Y} \in \mathbb{R}^N$. The attacker passes them through an equivalent global model and compares the resulting gradient update $\Delta^{(t)}(\mathbf{X}, \mathbf{Y}) = \sum_{u=0}^{\tau-1}\sum_{l=1}^{N}\nabla\ell(\mathbf{W}^{(t,u)};\mathbf{X}_l,\mathbf{Y}_l)$ to the target gradient $\nabla_\text{avg}^{(t)}$, where the singly indexed $\mathbf{X}_l$ and $\mathbf{Y}_l$ are the $l$th dummy image and label for the combined dataset. Following the gradient inversion framework, the goal is to iteratively refine $\mathbf{X}$ until it closely approximates the data used to compute the target gradient. The attacker solves the following optimization problem:

$$\hat{\mathbf{X}} = \arg\min_{\mathbf{X}}\left\|\Delta^{(t)}(\mathbf{X}, \mathbf{Y}) - \nabla_\text{avg}^{(t)}\right\|^2, \qquad (3)$$

where the evolving global model $\{\mathbf{W}^{(t,u)}\}_{u=0}^{\tau-1}$ requires only the knowledge of the total number of images, eliminating the need to know the number of clients and the image counts from all clients.

Figure 2: Pixel-level correct images reconstructed by the proposed attack before and after direct postprocessing (Section 3.2). The second and third rows show postprocessed and raw reconstructed images. The raw reconstruction results from our attack are low-resolution and have significant color aliasing. Our direct postprocessing method increases the resolution and simultaneously denoises the images while maintaining pixel-level correctness, revealing image details that make the reconstructions easier to recognize.

Finally, the attacker applies a postprocessing function $\phi(\cdot)$ to improve the quality of the reconstructed images $\tilde{\mathbf{X}} = \phi(\hat{\mathbf{X}})$. This shows that a curious client attacker is able to follow an approach similar to server-side gradient inversion and obtain reconstructed data from all other clients from only two consecutive versions of the model weights. We describe below two methods of postprocessing $\hat{\mathbf{X}}$ to improve its quality at either a pixel or semantic level.

## 3.2 DIRECT POSTPROCESSING

To reconstruct the target data more effectively, we introduce a direct postprocessing method that utilizes pretrained diffusion models to perform super resolution and denoising on the raw image reconstructions. The raw reconstructed images from the attack may be low-resolution or have pixel artifacts due to imperfect gradient inversion. This problem may also be more severe in our attack compared to server-side gradient inversion as the target gradient contains diluted information from multiple clients. To address this problem, we introduce a postprocessing implementation, $\phi(\cdot) \equiv \phi_{\text{direct}}(\cdot)$ that uses pretrained diffusion models to directly postprocess the raw reconstructed images. Diffusion models have demonstrated good performance in image generation and restoration tasks and are able to produce more realistic images with a lower likelihood of hallucination (Dhariwal & Nichol, 2021). Our method follows the denoising diffusion restoration models (DDRM) framework and utilizes a pretrained DDPM (Ho et al., 2020) as a backbone model. DDRM has demonstrated strong performance across various image restoration tasks, including super resolution and denoising (Kawar et al., 2022). By increasing resolution and removing noise, we aim to accurately reveal details of the ground truth images and make the reconstructions more recognizable.

## 3.3 RECONSTRUCTION AT A SEMANTIC LEVEL

In this subsection, we also introduce a method to reconstruct target images at the semantic level, $\phi(\cdot) \equiv \phi_{\text{semantic}}(\cdot)$, the masked diffusion enhancer (MDE). The goal of MDE is to generate sharp, high-resolution images from the low-resolution, color-aliased raw attack results. This approach complements the direct postprocessing technique, as the generated images resemble the raw reconstructions at the semantic level, rather than at the pixel level. The generated images preserve the shape and perceptual features of the target image while achieving photorealistic quality.

**Backbone Model.** We use masked diffusion transformer (MDT) as the backbone of our reconstruction technique. MDT has been proven to achieve state-of-the-art performance in image generation (Gao et al., 2023). Due to its extensive training and flexibility, MDT has learned a complex representation of each image class that enables it to accurately reconstruct each image's semantic features through projection onto the manifold. Following the diffusion framework, MDT generates images by starting from a Gaussian noise vector $\mathbf{X}_T \sim \mathcal{N}(\mathbf{0}, \mathbf{I})$, where $T$ is the total number of diffusion steps.

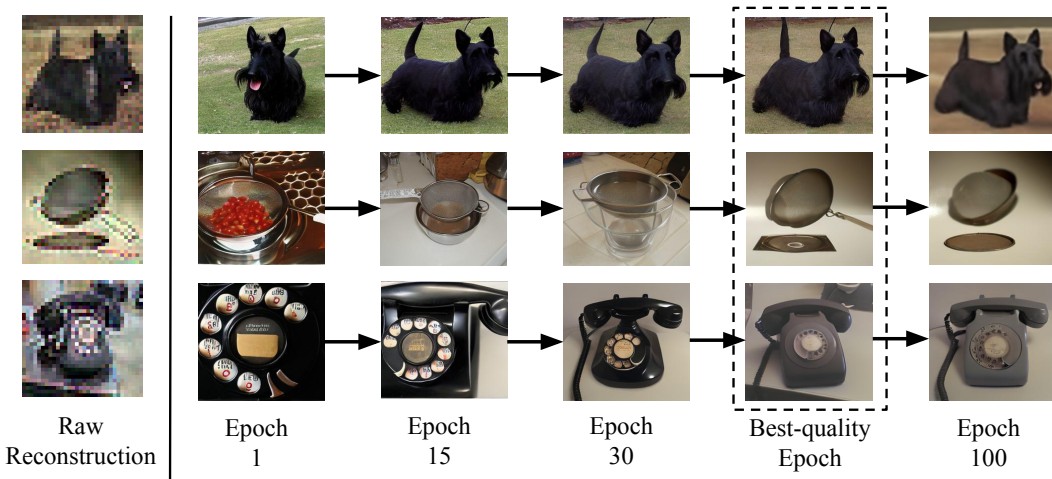

Figure 3: Photorealistic images reconstructed by the proposed semantic reconstruction method MDE. This diffusion-based method iteratively refines its generated image by referring to the raw reconstruction. As iterations progress, the image increasingly assumes the shape and perceptual features of the raw reconstruction. After some optimal epoch number (determined by visual inspection of the attacker), the reconstructed image strongly resembles the target at a high quality. Beyond this point, further optimization may produce blurry images due to overfitting.

At each step $t$, the model predicts a noise residual $\epsilon_\theta(\mathbf{X}_t)$, and uses it to refine the noisy image $\mathbf{X}_t$ to $\mathbf{X}_{t-1}$. After $T$ iterations, the initial noise vector $\mathbf{X}_T$ will be transformed into a high quality image $\mathbf{X}_0$. For our reconstruction technique, we leverage a pretrained MDT and freeze its model parameters throughout the process to maintain consistency in the image generation pipeline.

**Proposed Masked Diffusion Enhancer (MDE).** MDE optimizes the initial noise vector $\mathbf{X}_T$ to generate an image that closely matches a target image $\hat{\mathbf{X}}$. During optimization, the noise predictions $\epsilon_\theta(\mathbf{X}_t)$ at each timestep are treated as constants. The objective of MDE is to minimize the mean squared error (MSE) between the final generated image $\mu_\theta(X_T, T)$ and the target image $\hat{\mathbf{X}}$:

$$\tilde{\mathbf{X}}_T = \arg\min_{\mathbf{X}_T} \left\| \mu_\theta(\mathbf{X}_T, T) - \hat{\mathbf{X}} \right\|_2^2, \tag{4}$$

where $\mu_\theta(\mathbf{X}_T, T)$ denotes the final image produced from the initial noise vector $\mathbf{X}_T$ after all diffusion steps. By optimizing $\mathbf{X}_T$ based on the loss term, we guide the model to generate images that have the same shape and perceptual features as the target image.

## 4 EXPERIMENTAL RESULTS

This section first presents the performance of the proposed reconstruction attack in terms of image reconstruction quality against gradients averaged from multiple clients. Factors affecting reconstruction quality, including the number of local iterations and client batch size, will be analyzed. The postprocessing modules will be ablated to examine their benefits on image reconstruction. The state of the art will be compared and the limitation of the proposed attack will be discussed.

**Experimental Conditions.** We evaluate our reconstruction attacks using the ImageNet (Deng et al., 2009) and MNIST (LeCun et al., 1998) datasets. We employ LeNet (LeCun et al., 1998) and ResNet (He et al., 2016) as the global models and conduct experiments under the FedAvg framework. We vary the client batch size and the number of local training iterations. Each client computes local updates on the same number of training images, though this does not significantly affect the image reconstruction quality, as we verify in Appendix D. As more clients participate in training, the training image count from the global model's perspective increases proportionally. The attacker uses a learning rate of 0.1 to optimize the dummy data and the attack is conducted after the first FL round, following the approach of Yue et al. (2023). Before inverting the target gradient, the attacker encodes its dummy

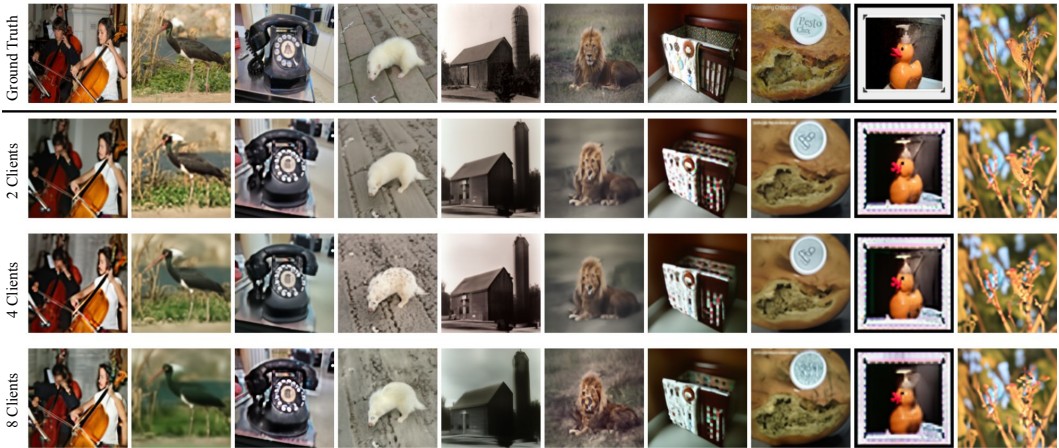

Figure 4: Pixel-level correct images reconstructed from the proposed honest-but-curious client-based attack. Rows 2–4 show reconstructions from gradients averaged across 2, 4, and 8 clients using LeNet5 as the global model with 16 images per client and 3 local training iterations. The images remain high quality even when the attack is performed against multiple clients.

data through bicubic sampling with a scale factor of 4 to reduce the number of unknown parameters. This has been proven to save convergence time and improve image quality in gradient inversion (Yue et al., 2023). We use LPIPS (Zhang et al., 2018) as the primary metric to evaluate quality of the attack's reconstructed images as it provides the best representation of perceptual image quality based on our experiments, through we observe similar trends for SSIM (Wang et al., 2004) and PSNR/MSE.

**Main Results.** Figure 5 shows the image reconstruction quality across three model architectures, LeNet, ResNet9, and ResNet18, measured by LPIPS as the number of clients increases. Each client trains for three local iterations on a batch of 16 images. As the number of clients increases, the total number of images from the attacker's perspective increases proportionally, up to 512 total images with 32 clients. We observe that reconstruction quality declines gradually as the number of clients and images increases, and with larger models. Figure 4 provides qualitative examples with LeNet for systems of 2-8 clients trained with three local iterations. The results show that while the reconstructions become noisier and exhibit stronger aliasing as the number of clients increases, the reconstructed images remain recognizable. Additional examples are provided in Appendix H and confirm that many reconstructions remain recognizable with 16 or more

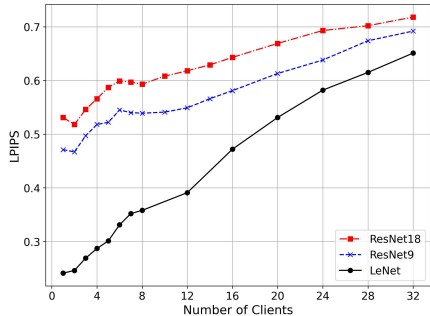

Figure 5: LPIPS of reconstructed images vs. number of clients with three different models: LeNet5, ResNet9, and ResNet18.

clients and that this trend holds across multiple models, batch sizes up to 128 images per client, and at least eight local training iterations. We note that clients follow the FedAvg protocol, though this is equivalent to FedSGD when only a single local iteration is used. In the FedSGD case, client and server-side gradient inversion are equivalent under the super-client approximation except for the number of images inverted, as we prove in Appendix A.

We further evaluate the impact of diffusion-based postprocessing on reconstruction quality. Figure 2 illustrates that DDPM-based direct postprocessing improves the raw reconstructions by simultaneously denoising and upsampling from $32 \times 32$ to $128 \times 128$ resolution. The raw reconstructions are constrained by the dummy data's encoding and often contain pixel artifacts and color aliasing due to imperfect inversion. Direct postprocessing addresses these issues, producing sharper images that are more recognizable and retain details closely matching the ground truth. Figure 3 shows the complementary effect of the masked diffusion enhancer (MDE), which refines the raw reconstructions

at the semantic level. MDE generates high-resolution images that preserve the geometric structure and perceptual features of the targets with photorealistic quality. While prolonged optimization may lead to blurring, at a manually-selected optimal point, the generated images closely resemble the raw reconstructions in both shape and semantics while achieving photorealistic quality.

**Factor Analysis.** Figure 6 reveals the effect of local iterations and client batch size on image reconstruction quality. These factors directly influence the attack's ability to invert the target gradient. As shown in the left plot of Figure 6, larger client batch sizes lead to worse reconstruction quality as the total number of images increases from the attacker's perspective. This problem is worse for client-side inversion as the total number of images from the attacker's perspective is a product of the client batch size and the number of participating clients, rather than simply the client batch size. Smaller batches add variability to updates, making them more informative for the attacker, whereas larger batches

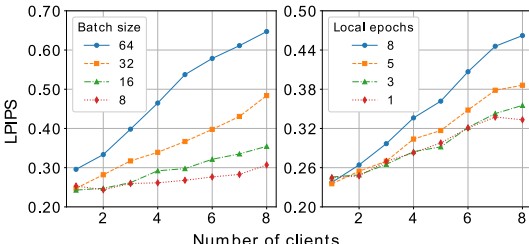

Figure 6: LPIPS of reconstructed images vs. number of clients, client batch size, and local iterations. Reconstruction quality worsens with more clients and larger batch sizes.

smooth updates and reduce the amount of exploitable information. The right plot of Figure 6 shows that increasing the number of local iterations leads to worse reconstructions when the number of clients is large, which is consistent with our theoretical result in Appendix A that the attacker's super-client approximation becomes worse when clients train for more local iterations. This is particularly important because federated learning often uses more local iterations to reduce communication. We further analyze the effect of more local iterations on the proposed attack in Appendix F.

*Lemma 1 (informal): We claim that the divergence between the attacker's single-client approximation and the true global update rule increases quadratically with the number of local iterations.* We provide a formal proof in Appendix A.

**Ablation Study.** We examine how much impact the direct and semantic postprocessing blocks have on the quality of the attacker's reconstructed images. Figure 7 shows that directly postprocessing the raw reconstructed images results in a 20–30% improvement in LPIPS for systems with 2–8 clients. Figure 2 visually compares reconstructed images from a system with four clients before and after direct postprocessing. The raw images are low resolution and may be somewhat difficult to recognize while the postprocessed versions show much finer details and have recognizable features. This demonstrates the utility of our direct postprocessing technique in increasing the pixel-wise accuracy and recognizability of the reconstructed images.

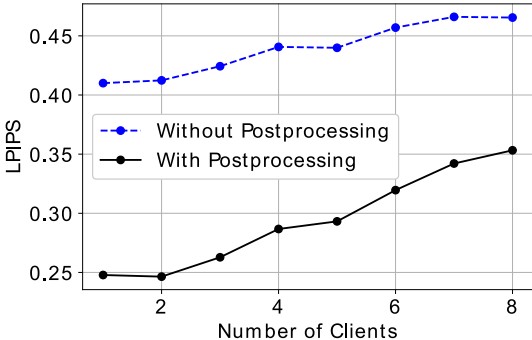

Figure 7: LPIPS of reconstructed images with varying number of clients. Our direct postprocessing technique significantly improves reconstruction quality compared to the raw attack results.

Additionally, Figure 3 shows the effect of postprocessing the raw reconstructed images using the proposed MDE. The final results have the same shape and perceptual features of the raw reconstructions without any pixel artifacts, color aliasing, or blurriness. However, MDE's goal is not to achieve pixel-wise accuracy so the generated images should not be compared quantitatively to the raw reconstructions.

**Comparison to ROG and FedInverse.** We compare our attack method to FedInverse (Wu et al., 2024) and our postprocessing modules to reconstruction from obfuscated gradients (ROG) (Yue et al., 2023). To the best of our knowledge, FedInverse is the only prior work addressing honest-but-curious client attacks, though it differs fundamentally from our setting. We do not compare our results with those from server-side attacks as client-side data reconstruction is a more difficult problem.

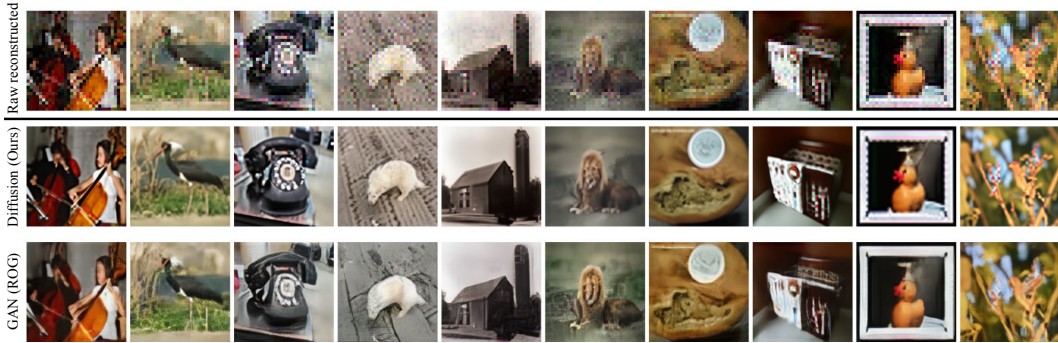

Figure 8: Pixel-level-correct reconstructed images using (row 2) our direct postprocessor and (row 3) ROG GAN (Yue et al., 2023). Our direct postprocessing method produces reconstructions that are more recognizable and have more accurate image details compared to the ROG GAN. However, our method also results in worse LPIPS and SSIM due to blurring.

Unlike our approach, which reconstructs data from gradients, FedInverse applies model inversion at late training stages to recover class-prototype images (the typical example of class $c$), rather than individual training samples. This difference indicates that the two methods are complementary rather than directly comparable. FedInverse is more general in scope, working with large, well-trained models, while our attack is most effective in earlier stages or with simpler models where gradient signals are larger. Because the attack surfaces differ, a direct numerical comparison would not be meaningful. Figure 9 illustrates the qualitative distinction in the privacy risks posed by each attack.

We compare reconstruction quality from our direct and semantic postprocessing techniques to the state-of-the-art postprocessing results achieved by Yue et al. (2023). Figure 8 shows that our technique generates images that are more recognizable but often blurry because of uncertainty in the fine image details. LPIPS is designed to evaluate image quality, rather than detail accuracy, and penalizes blurriness and pixelation much more than hallucination. We find that our direct postprocessing method yields reconstructions with higher LPIPS values than the state-of-the-art GAN-based postprocessing technique, though it achieves comparable SSIM, which is less sensitive to blurriness than LPIPS. Numerical results are provided in Appendix G.

Figure 9: (a) Evaluation on MNIST of the attack with MDE postprocessing compared to (b) the model inversion (MI) and model inversion with Hilbert–Schmidt independence criterion (MI-HSIC) approaches [reproduced from Wu et al. (2024)]. Only 5 examples were provided for each method. FedInverse generates class-prototype images while our attack breaches privacy at the instance level.

## 5 CONCLUSIONS AND FUTURE WORK

We have demonstrated that a curious client attacker can successfully reconstruct high-quality images from a small number of clients simply by participating in two consecutive training rounds. This type of attack does not alter the training process or introduce corrupted data, making it difficult to detect by the server or other clients in the system. Our findings indicate that the attack is highly effective when the number of participating clients is small or the available training examples are limited. This raises a significant concern for cross-silo FL, where participants collaborate specifically to overcome data scarcity (Li et al., 2020a). In such settings, our findings reveal a serious privacy risk, as the reconstruction of sensitive data becomes more feasible. Further research is needed to assess the robustness of more advanced FL frameworks against curious client attacks and develop effective defenses to preserve data privacy in cross-silo FL.

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

# A   THEORETICAL ANALYSIS OF DIVERGENCE BETWEEN TRUE GRADIENT UPDATE AND THAT FROM A SINGLE SUPER-CLIENT ATTACKER

We mathematically analyze the expected difference between the true gradient update and the attacker's single super-client approximation as a function of the number of local iterations $\tau$ and other hyperparameters of FedAvg.

> **Lemma 1.**   The expected difference between the FedAvg global weight/update and the single super-client approximated weight/update $\mathbb{E}[\delta(\tau)]$ is proportional to
>
> $$\eta_g^2 \tau(\tau - 1)(K - 1)/N,$$
>
> where $\eta_g$ is the learning rate, $\tau$ is the number of local iterations, $K$ is the number of clients, and $N$ is the number of training examples from all clients.

**Remark 1.** As the clients train beyond the first local iteration, their starting points for computing gradients increasingly diverge, worsening the attacker's super-client approximation.

**Remark 2.** When the number of clients $K$ is held constant, increasing the total number of training images $N$ reduces the expected divergence between the true FedAvg update and the attacker's single super-client approximation. This effect arises because each client's update becomes more stable and closer to the global average when computed over more examples, making the attacker's approximation more accurate.

**Sketch of the proof.** We first demonstrate the cases for $\tau = 2$ and 3, then guess the general form for $\tau = n$, and finally prove the case for $\tau = n + 1$ via mathematical induction. To simplify our analysis, we make the assumptions that image-label pairs $(X_{k,i}, Y_{k,i})$ are IID across client indices $k \in \{1, \ldots, K\}$, and that all clients use the same batch size $N_k = N/K, \forall k$.

**Proof:** We begin by formally defining the variables and expressions used within our analysis.

**Definition 1.** We define $g(W; Z_{k,i}) = \nabla\ell(W; Z_{k,i}) \in \mathbb{R}^{M \times 1}$ be the gradient of the training loss with respect to the model weights $W \in \mathbb{R}^{M \times 1}$, where $Z_{k,i} = (X_{k,i}, Y_{k,i})$ represents a single training image-label pair indexed by client $k$ and image number $i$.

**Definition 2.** We further define $J(W^{(t)}; Z_{k,i}) \in \mathbb{R}^{M \times M}$ as the Jacobian of $g(W; Z_{k,i})$ with respect to $W^{(t)}$, the model weights of the most recent global update. In our experiments, the attacker inverts the gradient of an untrained model so we substitute $W^{(t)} = W^0$, where $W^0 \in \mathbb{R}^{M \times 1}$ is the randomly-initialized weight vector before model training.

Recall in Section 3.1, we defined the update operations for FedAvg as follows:

$$\textbf{Local updates:} \quad W_k^{(t,\tau)} = W^{(t)} - \frac{\eta_g}{N_k} \sum_{u=0}^{\tau-1} \sum_{i=1}^{N_k} \nabla\ell(W_k^{(t,u)}; Z_{k,i}), \quad \forall k, \tag{5a}$$

$$\textbf{Global update:} \quad W_{\text{fed}}^{(t+1)} = \frac{1}{N} \sum_{k=1}^{K} N_k W_k^{(t,\tau)}. \tag{5b}$$

The definitions in Section 3.1 also allow us to explicitly denote the true global weight update as follows:

$$\textbf{True update:} \quad \eta_g^{(t)} \nabla_{\text{avg}}^{(t)} \overset{\text{def}}{=} W_{\text{fed}}^{(t+1)} - W^{(t)} = -\frac{\eta_g}{N} \sum_{k=1}^{K} \sum_{u=0}^{\tau-1} \sum_{i=1}^{N_k} \nabla\ell(W_k^{(t,u)}; Z_{k,i}). \tag{6}$$

**Reconstruction attack via a single super-client approximation.** The attacker will invert the difference between two sets of global model weights, $W^{(t)}$ and $W^{(t+1)}$, by using a single-client

approximation of the true update rule, which is defined as:

$$\textbf{Approximated update:} \quad \eta_g \Delta^{(t)} \overset{\text{def}}{=} W_{\text{single}}^{(t+1)} - W^{(t)} = -\frac{\eta_{\text{g}}}{N} \sum_{u=0}^{\tau-1} \sum_{i=1}^{N} \nabla \ell(W^{(t,u)}; Z_{k,i}) \quad (7a)$$

$$= -\frac{\eta_{\text{g}}}{N} \sum_{k=1}^{K} \sum_{u=0}^{\tau-1} \sum_{i=1}^{N_k} \nabla \ell(W^{a(t,u)}; Z_{k,i}), \quad (7b)$$

where $W^a$ represents the placeholder variables for weights that the curious client attacker aims to estimate. We explore the simplified scenario in which the client correctly estimated all $Z_{k,i}$s.
**Divergence measure.** Given that the attacker's approximation of the global update differs from the true update rule, we define a measure, $\delta(\tau)$, to quantify the divergence between the attacker's single-client approximation and the true model update rule for $K$ clients as follows:

$$\delta(\tau) \overset{\text{def}}{=} \eta_g^{(t)} \nabla_{\text{avg}}^{(t)} - \eta_g \Delta^{(t)} = W_{\text{fed}}^{(t+1)} - W_{\text{single}}^{(t+1)} \quad (8a)$$

$$= -\frac{\eta_{\text{g}}}{N} \sum_{k=1}^{K} \sum_{u=0}^{\tau-1} \sum_{i=1}^{N_k} \left[ g(W_k^{(t,u)}; Z_{k,i}) - g(W^{a(t,u)}; Z_{k,i}) \right]. \quad (8b)$$

Applying a first-order Taylor expansion around $W^0$, we obtain:

$$\delta(\tau) \approx -\frac{\eta_{\text{g}}}{N} \sum_{k=1}^{K} \sum_{u=0}^{\tau-1} \sum_{i=1}^{N_k} \left[ J(W^0; Z_{k,i})(W_k^{(t,u)} - W^{a(t,u)}) \right]. \quad (9)$$

With a slight misuse of notations, we will use equals signs instead of approximation signs for further equations involving the same Taylor expansion.

## A.1 Demonstration for $\tau \in \{1, 2, 3\}$

**When $\tau = 1$.** When the clients train for only one local iteration, the attacker's approximation is equivalent to the true model update:

$$\delta(1) = -\frac{\eta_{\text{g}}}{N} \sum_{k=1}^{K} \sum_{i=1}^{N_k} \left[ J(W^0; Z_{k,i})(W_k^{(t,0)} - W^{a(t,0)}) \right] = 0, \quad (10)$$

because $W_k^{(t,0)} = W^{a(t,0)} = W^0$. **When $\tau = 2$.** When the clients train for multiple local iterations, we may obtain a recursive relationship between $\delta(n)$ and $\delta(n-1)$ from equation 9. For $\delta(2)$ and $\delta(1)$, this yields:

$$\delta(2) = \delta(1) - \frac{\eta_g}{N} \sum_{k=1}^{K} \sum_{i=1}^{N_k} J\left(W_{k,i}; Z_{k,i}\right) \left(W_k^{(t,1)} - W^{a(t,1)}\right). \quad (11)$$

We study the divergence in the expectation sense as follows:

$$\mathbb{E}\left[\delta(2)\right] = 0 - \frac{\eta_g}{N} \sum_{k=1}^{K} \sum_{i=1}^{N_k} \mathbb{E}\left[ J\left(W^0; Z_{k,i}\right) \left(W_k^{(t,1)} - W^{a(t,1)}\right) \right] \quad (12a)$$

$$= -\frac{\eta_g}{N} \sum_{k=1}^{K} \sum_{i=1}^{N_k} \mathbb{E}\left\{ \underbrace{\mathbb{E}\left[ J\left(W^0; Z_{k,i}\right) \left(W_k^{(t,1)} - W^{a(t,1)}\right) | W^0 \right]}_{\overset{\text{def}}{=} h_{k,i}^{(2)}(W^0)} \right\}, \quad (12b)$$

where $W^0$ is the randomly initialized weight vector before model training. Substituting the local update rule equation 5a, and the attacker's update rule equation 7b into $h_{k,i}^{(2)}(W^0)$, we obtain:

$$h_{k,i}^{(2)}(W^0) = \mathbb{E}\left[ J\left(W^0; Z_{k,i}\right) \left( W^0 - \frac{\eta_g}{N_k} \sum_{i=1}^{N_k} g\left(W^0; Z_{k,i}\right) - W^0 \right. \right. \tag{13a}$$

$$\left. \left. + \frac{\eta_g}{N} \sum_{k=1}^{K} \sum_{i=1}^{N_k} g\left(W^0; Z_{k,i}\right) \right) \middle| W^0 \right]$$

$$= \eta_g \mathbb{E}\left[ J\left(W^0; Z_{k,i}\right) \left( \left( \frac{1}{N} - \frac{1}{N_k} \right) \sum_{i=1}^{N_k} g\left(W^0; Z_{k,i}\right) \right. \right. \tag{13b}$$

$$\left. \left. + \frac{1}{N} \sum_{\substack{k_1=1 \\ k_1 \neq k}}^{K} \sum_{i=1}^{N_k} g\left(W^0; Z_{k_1,i}\right) \right) \middle| W^0 \right]$$

$$h_{k,i}^{(2)}(W^0) = \eta_g \left( \frac{1}{N} - \frac{1}{N_k} \right) \mathbb{E}\left[ J\left(W^0; Z_{k,i}\right) \sum_{i=1}^{N_k} g\left(W^0; Z_{k,i}\right) \middle| W^0 \right]$$

$$+ \frac{\eta_g}{N} \mathbb{E}\left[ J\left(W^0; Z_{k,i}\right) \middle| W^0 \right] \mathbb{E}\left[ \sum_{\substack{k_1=1 \\ k_1 \neq k}}^{K} \sum_{i=1}^{N_k} g\left(W^0; Z_{k_1,i}\right) \middle| W^0 \right] \tag{14}$$

$$h_{k,i}^{(2)}(W^0) = \eta_g \left( \frac{1}{N} - \frac{1}{N_k} \right) \left\{ \mathbb{E}\left[ J\left(W^0; Z_{k,i}\right) g\left(W^0; Z_{k,i}\right) \middle| W^0 \right] \right.$$

$$\left. + \sum_{\substack{i'=1 \\ i' \neq i}}^{N_k} \mathbb{E}\left[ J\left(W^0; Z_{k,i}\right) \middle| W^0 \right] \mathbb{E}\left[ g\left(W^0; Z_{k,i_1}\right) \middle| W^0 \right] \right\} \tag{15}$$

$$+ \frac{\eta_g}{N} \mathbb{E}\left[ J\left(W^0; Z_{k,i}\right) \middle| W^0 \right] \mathbb{E}\left[ \sum_{\substack{k_1=1 \\ k_1 \neq k}}^{K} \sum_{i=1}^{N_k} g\left(W^0; Z_{k,i}\right) \middle| W^0 \right],$$

where equation 14 is due to the independence between training examples $Z_{k,i}$ and $\{Z_{k_1,i}\}_{k_1 \neq k}$ and equation 15 is due to the independence between training examples $Z_{k,i}$ and $\{Z_{k,i_1}\}_{i_1 \neq i}$. Let us define

$$\mu_g = \mathbb{E}\left[ g\left(W^0; Z_{k,i}\right) \middle| W^0 \right], \tag{16a}$$

$$\mu_J = \mathbb{E}\left[ J\left(W^0; Z_{k,i}\right) \middle| W^0 \right]. \tag{16b}$$

Substituting these expressions into equation 15, we obtain:

$$h_{k,i}^{(2)}(W^0) = \frac{\eta_g}{N} \mu_J (N - N_k) \mu_g + \eta_g \left( \frac{1}{N} - \frac{1}{N_k} \right) \left\{ \mathbb{E}\left[ J \cdot g \mid W^0 \right] + (N_k - 1) \mu_g \mu_J \right\} \tag{17a}$$

$$= \eta_g \cdot \frac{1-K}{N} \left\{ \mathbb{E}\left[ J \cdot g \mid W^0 \right] - \mu_J \cdot \mu_g \right\} \tag{17b}$$

$$= \eta_g \cdot \frac{1-K}{N} \cdot \mathrm{Cov}(J, g \mid W^0), \tag{17c}$$

where

$$\left[ \mathrm{Cov}(X, Y) \right]_{i,j} := \sum_{k=1}^{p} \mathrm{Cov}(x_{i,k}, y_{k,j}), \tag{18}$$

for $X = [x_{i,k}]_{m \times p}$ and $Y = [y_{k,j}]_{p \times n}$. Substituting equation 17c back into equation 12b, we obtain:

$$\mathbb{E}\left[\delta(2)\right] = -\frac{\eta_g}{N} \sum_{k=1}^{K} \sum_{i=1}^{N_k} h_{k,i}^{(2)}(W^0) \tag{19a}$$

$$= \eta_g^2 \cdot \frac{K-1}{N} \cdot \underbrace{\mathbb{E}\left[\text{Cov}(J, g \mid W^0)\right]}_{\stackrel{\text{def}}{=} \overline{\text{Cov}}(J,g)}. \tag{19b}$$

From this result, we can see that the divergence for two local iterations is nonzero and increases linearly with the number of clients $K$ when the total number of images $N$ is fixed.

**When $\tau = 3$.** Similar to $\tau = 2$, we have the recursive relationship between $\delta(3)$ and $\delta(2)$:

$$\delta(3) = \delta(2) - \frac{\eta_g}{N} \sum_{k=1}^{K} \sum_{i=1}^{N_k} J\left(W_{k,i}; Z_{k,i}\right)\left(W_k^{(t,2)} - W^{a(t,2)}\right). \tag{20}$$

Taking the expected value, we then obtain:

$$\mathbb{E}\left[\delta(3)\right] = \mathbb{E}\left[\delta(2)\right] - \frac{\eta_g}{N} \sum_{k=1}^{K} \sum_{i=1}^{N_k} \mathbb{E}\left\{\underbrace{\mathbb{E}\left[J\left(W^0; Z_{k,i}\right)\left(W_k^{(t,2)} - W^{a(t,2)}\right) \mid W^0\right]}_{\stackrel{\text{def}}{=} h_{k,i}^{(3)}(W^0)}\right\}. \tag{21}$$

Substituting the local update rule equation 5a, and the attacker's update rule equation 7b into $h_{k,i}^{(3)}(W^0)$, we obtain:

$$h_{k,i}^{(3)} = \mathbb{E}\left[J\left(W^0; Z_{k,i}\right)\left(W_k^{(t,2)} - W^{a(t,2)}\right)\Big| W^0\right] \tag{22a}$$

$$= \mathbb{E}\Bigg[J\left(W^0; Z_{k,i}\right)\Bigg(W^0 - \frac{\eta_g}{N_k}\sum_{i=1}^{N_k} g\left(W^0; Z_{k,i}\right) - \frac{\eta_g}{N_k}\sum_{k=1}^{N_k} g(W^0; Z_{k,i})$$

$$- \frac{\eta_g}{N_k}\sum_{i=1}^{N_k} \nabla g\left(W^0; Z_{k,i}\right)\left(W^{(t,1)} - W^0\right) - W^0 + \frac{\eta_g}{N}\sum_{k=1}^{K}\sum_{i=1}^{N_k} g\left(W^0; Z_{k,i}\right)$$

$$+ \frac{\eta_g}{N}\sum_{k=1}^{K}\sum_{i=1}^{N_k} g\left(W^0; Z_{k,i}\right) + \frac{\eta_g}{N}\sum_{k=1}^{K}\sum_{i=1}^{N_k} J\left(W^0; Z_{k,i}\right)\left(W^{(t,1)} - W^0\right)\Bigg)\Bigg| W^0\Bigg]. \tag{22b}$$

Distributing the expectation, we obtain:

$$h_{k,i}^{(3)} = 2\mathbb{E}\left[J\left(W^0; Z_{k,i}\right)\left(W_k^{(t,1)} - W^{a(t,1)}\right)\right]$$

$$- \frac{\eta_g^2}{N^2}\mathbb{E}\left[J\left(W^0; Z_{k,i}\right)\sum_{k_1=1}^{K}\sum_{i_1=1}^{N_k}\sum_{k_2=1}^{K}\sum_{i_2=1}^{N_k} J\left(W^0; Z_{k_1,i_1}\right) g\left(W^0; Z_{k_2,i_2}\right)\right]$$

$$+ \frac{\eta_g^2}{N_k^2}\mathbb{E}\left[J\left(W^0; Z_{k,i}\right)\sum_{i_1=1}^{N_k}\sum_{i_2=1}^{N_k} J\left(W^0; Z_{k,i_1}\right) g\left(W^0; Z_{k,i_2}\right)\right]. \tag{23}$$

We note that first term of equation 23 mirrors the case where $\tau = 2$. The summations in the second and third terms can each be divided into four parts based on the indices of the summations:

1. $(k_1, i_1) = (k, i)$, and $(k_2, i_2) = (k, i)$.

2. $(k_1, i_1) = (k, i)$, and $(k_2, i_2) \neq (k, i)$.

3. $(k_1, i_1) \neq (k, i)$, and $(k_2, i_2) = (k, i)$.

4. $(k_1, i_1) \neq (k, i)$, and $(k_2, i_2) \neq (k, i)$.

Equal indices indicate two terms depend on the same $Z_{k,i}$, while unequal indices indicate the terms depend on different $Z_{k,i}$s and are independent under our assumption that the training images are IID. Splitting the terms, we obtain:

$$
\begin{aligned}
h_{k,i}^{(3)} =\ & 2\mathbb{E}\left[J\left(W^0; Z_{k,i}\right)\left(W_k^{(t,1)} - W^{a(t,1)}\right)\middle| W^0\right] \\
& - \frac{\eta_g^2}{N^2}\mathbb{E}\left[J\left(W^0; Z_{k,i}\right)\left(J\left(W^0; Z_{k,i}\right)g\left(W^0; Z_{k,i}\right) + J\left(W^0; Z_{k,i}\right)\sum_{k_1=1}^{K}\sum_{i_1=1}^{N_k} g\left(W^0; Z_{k_1,i_1}\right)\right.\right. \\
& \left.\left. + \sum_{k_1=1}^{K}\sum_{i_1=1}^{N_k} J\left(W^0; Z_{k_1,i_1}\right)g\left(W^0; Z_{k,i}\right) + \sum_{k_1=1}^{K}\sum_{i_1=1}^{N_k}\sum_{k_2=1}^{K}\sum_{i_2=1}^{N_k} J\left(W^0; Z_{k_1,i_1}\right)g\left(W^0; Z_{k_2,i_2}\right)\right)\middle| W^0\right] \\
& + \frac{\eta_g^2}{N_k^2}\mathbb{E}\left[J\left(W^0; Z_{k,i}\right)\left(J\left(W^0; Z_{k,i}\right)g\left(W^0; Z_{k,i}\right) + J\left(W^0; Z_{k,i}\right)\sum_{i_1=1}^{N_k} g\left(W^0; Z_{k,i_1}\right)\right.\right. \\
& \left.\left. + \sum_{i_1=1}^{N_k} J\left(W^0; Z_{k,i_1}\right)g\left(W^0; Z_{k,i}\right) + \sum_{i_1=1}^{N_k}\sum_{i_2=1}^{N_k} J\left(W^0; Z_{k,i_1}\right)g\left(W^0; Z_{k,i_2}\right)\right)\middle| W^0\right].
\end{aligned} \tag{24}
$$

Given our assumption of independence among $Z_{k,i}$, we distribute the expectation operator:

$$
\begin{aligned}
h_{k,i}^{(3)} =\ & 2\mathbb{E}\left[J\left(W^0; Z_{k,i}\right)\left(W_k^{(t,1)} - W^{a(t,1)}\right)\middle| W^0\right] \\
& - \frac{\eta_g^2}{N^2}\left(\mathbb{E}\left[J(W^0; Z_{k,i})J\left(W^0; Z_{k,i}\right)g\left(W^0; Z_{k,i}\right)\middle| W^0\right]\right. \\
& + \mathbb{E}\left[J(W^0; Z_{k,i})J\left(W^0; Z_{k,i}\right)\middle| W^0\right]\mathbb{E}\left[\sum_{k_1=1}^{K}\sum_{i_1=1}^{N_k} g\left(W^0; Z_{k_1,i_1}\right)\middle| W^0\right] \\
& + \mathbb{E}\left[\sum_{k_1=1}^{K}\sum_{i_1=1}^{N_k} J\left(W^0; Z_{k_1,i_1}\right)\middle| W^0\right]\mathbb{E}\left[J(W^0; Z_{k,i})g\left(W^0; Z_{k,i}\right)\middle| W^0\right] \\
& \left. + \mathbb{E}\left[J(W^0; Z_{k,i})\middle| W^0\right]\mathbb{E}\left[\sum_{k_1=1}^{K}\sum_{i_1=1}^{N_k}\sum_{k_2=1}^{K}\sum_{i_2=1}^{N_k} J\left(W^0; Z_{k_1,i_1}\right)g\left(W^0; Z_{k_2,i_2}\right)\middle| W^0\right]\right) \\
& + \frac{\eta_g^2}{N_k^2}\left(\mathbb{E}\left[J(W^0; Z_{k,i})J\left(W^0; Z_{k,i}\right)g\left(W^0; Z_{k,i}\right)\middle| W^0\right]\right. \\
& + \mathbb{E}\left[J(W^0; Z_{k,i})J\left(W^0; Z_{k,i}\right)\middle| W^0\right]\mathbb{E}\left[\sum_{i_1=1}^{N_k} g\left(W^0; Z_{k,i_1}\right)\middle| W^0\right] \\
& + \mathbb{E}\left[\sum_{i_1=1}^{N_k} J\left(W^0; Z_{k,i_1}\right)\middle| W^0\right]\mathbb{E}\left[J(W^0; Z_{k,i})g\left(W^0; Z_{k,i}\right)\middle| W^0\right] \\
& \left. + \mathbb{E}\left[J(W^0; Z_{k,i})\middle| W^0\right]\mathbb{E}\left[\sum_{i_1=1}^{N_k}\sum_{i_2=1}^{N_k} J\left(W^0; Z_{k,i_1}\right)g\left(W^0; Z_{k,i_2}\right)\middle| W^0\right]\right).
\end{aligned} \tag{25}
$$

In addition to the already defined terms from our analysis on $\tau = 2$, we further define:

$$
J^2 = J\left(W^0; Z_{k,i}\right)J\left(W^0; Z_{k,i}\right), \tag{26a}
$$

$$
\mu_{J^2} = \mathbb{E}\left[J(W^0; Z_{k,i})J\left(W^0; Z_{k,i}\right)\right]. \tag{26b}
$$

Substituting them into equation 25, we obtain:

$$
\begin{aligned}
h_{k,i}^{(3)} =\ & 2\mathbb{E}\left[ J\left(W^0; Z_{k,i}\right)\left(W_k^{(t,1)} - W^{a(t,1)}\right)\Big|W^0\right] \\
& - \frac{\eta_g^2}{N_k^2}\left(\mathbb{E}(J^2 \cdot g \mid W^0) + \mu_{J^2}\mu_g(N-1) + \mu_J\mathbb{E}(J \cdot g \mid W^0)(N-1)\right. \\
& + \mu_J[(N-1)\mathbb{E}(J \cdot g \mid W^0) + (N^2 - 3N + 2)\mu_J\mu_g]) \\
& + \frac{\eta_g^2}{N_k^2}\left(\mathbb{E}(J^2 \cdot g \mid W^0) + \mu_{J^2}\mu_g(N_k-1) + \mu_J\mathbb{E}(J \cdot g \mid W^0)(N_k-1)\right) \\
& + \mu_J[(N_k-1)\mathbb{E}(J \cdot g \mid W^0) + (N_k^2 - 3N_k + 2)\mu_J\mu_g])
\end{aligned}
\tag{27}
$$

$$
\begin{aligned}
h_{k,i}^{(3)} =\ & 2\mathbb{E}\left[ J\left(W^0; Z_{k,i}\right)\left(W_k^{(t,1)} - W^{a(t,1)}\right)\Big|W^0\right] + \eta_g^2\bigg\{ \tfrac{K^2-1}{N^2}\cdot\mathrm{Cov}(J^2, g \mid W^0) \\
& - \tfrac{1-K}{N}\cdot\mathrm{Var}(J \mid W^0)\mu_g + \tfrac{2(K-1)(N-K-1)}{N^2}\cdot\mu_J\,\mathrm{Cov}(J, g \mid W^0)\bigg\},
\end{aligned}
\tag{28}
$$

where $\mathrm{Var}(X) = \mathrm{Cov}(X, Y)$ where $X = Y$. Substituting into the recursive form of $\delta(3)$ equation 21, we obtain:

$$
\begin{aligned}
\mathbb{E}\left[\delta(3)\right] = \mathbb{E}\left[\delta(2)\right] - \frac{\eta_g}{N}\sum_{k=1}^{K}\sum_{i=1}^{N_k}\mathbb{E}\bigg\{ & 2\mathbb{E}\left[ J\left(W^0; Z_{k,i}\right)\left(W_k^{(t,1)} - W^{a(t,1)}\right)\Big|W^0\right] \\
& + \eta_g^2\bigg[ \tfrac{K^2-1}{N^2}\cdot\mathrm{Cov}(J^2, g \mid W^0) - \tfrac{1-K}{N}\cdot\mathrm{Var}(J \mid W^0)\mu_g \\
& + \tfrac{2(K-1)(N-K-1)}{N^2}\cdot\mu_J\,\mathrm{Cov}(J, g \mid W^0)\bigg]\bigg\}.
\end{aligned}
\tag{29}
$$

Reusing the result from $\tau = 2$ equation 19b to simplify the first inner term, we obtain:

$$
\begin{aligned}
\mathbb{E}\left[\delta(3)\right] = 3\mathbb{E}\left[\delta(2)\right] + \eta_g^3\bigg[ & \tfrac{K-1}{N}\cdot\overline{\mathrm{Cov}}(J)\mu_g \\
& - \tfrac{K^2-1}{N^2}\cdot\overline{\mathrm{Cov}}(J^2, g) - \tfrac{2(K-1)(N-K-1)}{N^2}\cdot\mu_J\cdot\overline{\mathrm{Cov}}(J, g)\bigg].
\end{aligned}
\tag{30}
$$

The expression has two terms: one proportional to $\eta_g^2$ and another to $\eta_g^3$. In typical settings where $\eta_g$ is small (e.g. $\eta_g = 0.003$ in our experiments) and $N \gg K$ (e.g., $N = 64$, $K = 4$), the cubic term is suppressed by at least one additional order of magnitude relative to the quadratic term. The coefficients of $\eta_g^3$ involve averaged variance and covariance terms, which are unlikely to be large enough to counteract this suppression. Specifically, the covariance of $J^2$ and $g$ would need to be several orders of magnitude larger than the covariance of $J$ and $g$ to overcome the intrinsic $\eta_g^3$ downscaling.

Using our experimental observation $\mu_g \approx -0.004$ and our set learning rate of $\eta_g = 0.003$, we show that the cubic term is much smaller than the quadratic term under the range of $\overline{\mathrm{Cov}}(J)$, $\overline{\mathrm{Cov}}(J^2, g) \leq 100 \cdot \overline{\mathrm{Cov}}(J, g)$. This is based on the relationship between $J$ and $g$ and that the mean and standard deviation of $g$ are each experimentally determined to be close to zero. Under typical experimental conditions, we expect that these values would be much smaller than the given bound. However, even if $\overline{\mathrm{Cov}}(J, g) = 1$, $\overline{\mathrm{Cov}}(J) = \overline{\mathrm{Cov}}(J^2, g) = 100$, and $\mu_J = 1000$, the quadratic term is still 33.1 times greater than the cubic term. Given this, we conclude that the cubic term is negligible and exclude it from further analysis.

The expression then shows that the deviation with three local iterations is three times greater than the deviation with two local iterations.

## A.2 GENERAL FORM

For $\tau \in \{2, 3\}$, the divergence contains a factor of $\eta_g^2 \cdot \frac{K-1}{N} \cdot \overline{\mathrm{Cov}}(J, g)$, which dominates the expression as further terms are multiplied by higher powers of $\eta_g$, and $\eta_g$ is small. From equation 22b,

it is apparent that each local iteration beyond $\tau = 1$ will increase the factor of $W_k^{(t,1)} - W^{a(t,1)}$ by one and that the higher order terms of those iterations will have factors of $\eta_g^m, m > 2$. These higher order terms should be less than or equal to the case where $\tau = 3$ and are small relative to the first term. This pattern yields the following general form:

$$\mathbb{E}[\delta(n)] = \frac{n(n-1)}{2} \cdot \eta_g^2 \cdot \frac{K-1}{N} \cdot \overline{\text{Cov}}(J, g) + O, \tag{31}$$

where $O$ represents terms multiplied by $\eta_g^m$ where $m > 2$. We assume these terms are negligible based on our prior analysis.

**Mathematical Induction.** We prove by mathematical induction that equation 31 is the general form of $\mathbb{E}[\delta(n)]$. We first demonstrate the base case, where $\tau = 1$:

$$\delta(1) = \frac{1(1-1)}{2} \cdot \eta_g^2 \cdot \frac{K-1}{N} \cdot \overline{\text{Cov}}(J, g) = 0. \tag{32}$$

We now assume equation 31 is true and try to prove:

$$\mathbb{E}[\delta(n+1)] = \frac{(n+1)n}{2} \cdot \eta_g^2 \cdot \frac{K-1}{N} \cdot \overline{\text{Cov}}(J, g) + O. \tag{33}$$

From equation 9, we know that $\delta(n+1)$ is recursively related to $\delta(n)$ by the following expression:

$$\delta(n+1) = \delta(n) - \frac{\eta_g}{N} \sum_{k=1}^{K} \sum_{i=1}^{N_k} \left[ J(W^0; Z_{k,i})(W_k^{(t,n)} - W^{a(t,n)}) \right]. \tag{34}$$

Substituting the local update rule equation 5a and the attacker's update rule equation 7b, we obtain:

$$\delta(n+1) = \delta(n) - \frac{\eta_g}{N} \sum_{k=1}^{K} \sum_{i=1}^{N_k} \left[ J(W^0; Z_{k,i})\eta_g \left\{ \frac{1}{N} \sum_{k=1}^{K} \sum_{u=0}^{n-1} \sum_{i=1}^{N_k} g(W^{a(t,u)}; Z_{k,i}) \right. \right. \tag{35}$$
$$\left. \left. - \frac{1}{N_k} \sum_{u=0}^{n-1} \sum_{i=1}^{N_k} g(W_k^{(t,u)}; Z_{k,i}) \right\} \right].$$

We then apply the same Taylor expansion used for equation 9 to expand $g$.

$$\delta(n+1) = \delta(n) - \frac{\eta_g}{N} \sum_{k=1}^{K} \sum_{i=1}^{N_k} \left[ J(W^0; Z_{k,i})\eta_g \left\{ \frac{1}{N} \sum_{k=1}^{K} \sum_{u=0}^{n-1} \sum_{i=1}^{N_k} \left( g(W^0; Z_{k,i}) \right. \right. \right.$$
$$\left. + J(W^0; Z_{k,i}) \left[ W^{a(t,u)} - W^0 \right] \right) \tag{36}$$
$$\left. \left. - \frac{1}{N_k} \sum_{u=0}^{n-1} \sum_{i=1}^{N_k} \left( g(W^0; Z_{k,i}) + J(W^0; Z_{k,i}) \left[ W_k^{(t,u)} - W^0 \right] \right) \right\} \right]$$

$$\delta(n+1) = \delta(n) - \frac{\eta_g}{N} \sum_{k=1}^{K} \sum_{i=1}^{N_k} \left[ J(W^0; Z_{k,i}) \left\{ -n \cdot W^0 + n \cdot \frac{\eta_g}{N} \sum_{k=1}^{K} \sum_{i=1}^{N_k} g(W^0; Z_{k,i}) \right. \right.$$
$$- n \cdot \frac{\eta_g}{N_k} \sum_{i=1}^{N_k} g(W^0; Z_{k,i}) + \frac{\eta_g}{N} \sum_{k=1}^{K} \sum_{u=0}^{n-1} \sum_{i=1}^{N_k} J(W^0; Z_{k,i}) \left[ W^{a(t,u)} - W^0 \right] \tag{37}$$
$$\left. \left. + n \cdot W^0 - \frac{\eta_g}{N_k} \sum_{u=0}^{n-1} \sum_{i=1}^{N_k} J(W^0; Z_{k,i}) \left[ W_k^{(t,u)} - W^0 \right] \right\} \right]$$

We separate the terms, add $n \cdot W^0 - n \cdot W^0$ to the inner expression, and substitute equation 5a and equation 7b to expand the first-order terms.

$$
\begin{aligned}
\delta(n+1) = \ & \delta(n) - \frac{\eta_{\mathrm{g}}}{N} \sum_{k=1}^{K} \sum_{i=1}^{N_k} \left[ J(W^0; Z_{k,i}) \left\{ n \left( W_k^{(t,1)} - W^{a(t,1)} \right) \right. \right. \\
& + \frac{\eta_{\mathrm{g}}}{N} \sum_{k=1}^{K} \sum_{u=0}^{n-1} \sum_{i=1}^{N_k} J(W^0; Z_{k,i}) \left[ -\frac{\eta_{\mathrm{g}}}{N} \sum_{k=1}^{K} \sum_{u'=0}^{u-1} \sum_{i=1}^{N_k} g(W^{a(t,u')}; Z_{k,i}) \right] \\
& \left. \left. - \frac{\eta_{\mathrm{g}}}{N_k} \sum_{u=0}^{n-1} \sum_{i=1}^{N_k} J(W^0; Z_{k,i}) \left[ -\frac{\eta_{\mathrm{g}}}{N_k} \sum_{u'=0}^{u-1} \sum_{i=1}^{N_k} g(W_k^{(t,u')}; Z_{k,i}) \right] \right\} \right].
\end{aligned} \tag{38}
$$

We can see that the second and third terms within the outer summation will have a factor of $\eta_g^3$ or higher. Under our prior assumption, we determine that they will be small relative to the $\eta_g^2$ term and neglect them. We also note that for $\tau > 3$, the additional terms beyond those that appeared for $\tau = 3$ will have factors of $\eta_g$ raised to powers greater than three, making them much smaller than the terms we showed were negligible for $\tau = 3$. Taking the expectation and using our result from $\tau = 2$, we then rewrite the first term, obtaining:

$$
\mathbb{E}[\delta(n+1)] = \mathbb{E}[\delta(n)] + n \cdot \tfrac{K-1}{N} \cdot \overline{\mathrm{Cov}}(J, g) + O \tag{39a}
$$

$$
= \frac{n(n-1)}{2} \cdot \eta_g^2 \cdot \tfrac{K-1}{N} \cdot \overline{\mathrm{Cov}}(J, g) + O + n \cdot \tfrac{K-1}{N} \cdot \overline{\mathrm{Cov}}(J, g) + O \tag{39b}
$$

$$
= \frac{(n+1)n}{2} \cdot \eta_g^2 \cdot \tfrac{K-1}{N} \cdot \overline{\mathrm{Cov}}(J, g) + O. \tag{39c}
$$

This completes the proof for the general form equation 31 for the discrepancy between the attacker's single super-client approximation and the true global update.

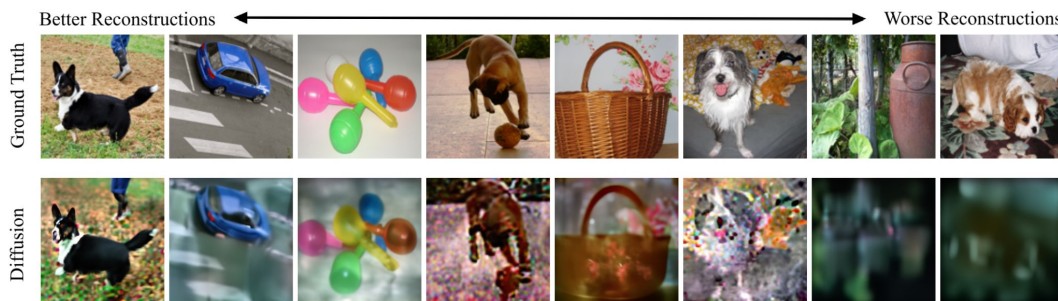

Figure 10: Pixel-level-correct reconstructed images from a system with 16 clients. With more clients, our reconstruction technique can reconstruct only a subset of training images with high quality, whereas others show distortion and color aliasing artifacts. Each client has 16 images and trains for 3 local iterations using LeNet5 as the model architecture.

## B DISCUSSION

**Limitations.** Our attack struggles to reconstruct high-quality images in systems where the number of clients is large. As the information contained within the attacker's target gradient is averaged from more clients, it becomes more difficult to reconstruct high-quality images. With more than 8 clients, we observe that some reconstructed images remain high quality while others exhibit significant degradation or are not recognizable. Figure 10 shows the varying quality in our reconstructed images in a system of 16 clients. Additionally, we observe that the postprocessors are often able to restore image details that may not be obvious to a human observer looking at the raw reconstruction results. However, they are not able to restore images when the raw reconstruction result does not provide enough information, which is a problem common to all postprocessing tasks.

**Unknown Learning Rate.** We assume in our experiments that the attacker knows the global learning rate $\eta_g$. This assumption simplifies the attack but need not be true for the attack to be effective. If the learning rate is applied globally (by the central server) and the attacker's guess differs from the true value, the target gradient will be inversely scaled by a factor of the ratio between the guessed learning rate and the true learning rate, leading to poor reconstruction quality. For simplicity, we set the base $\eta_g = 1$ and examine the impact on reconstruction when it is unknown to the attacker. The attacker uses its own training images to evaluate reconstruction quality as it knows they will be in the set of reconstructed images. Figure 11 shows that reconstruction quality degrades rapidly as the guessed and true learning rates diverge. How-

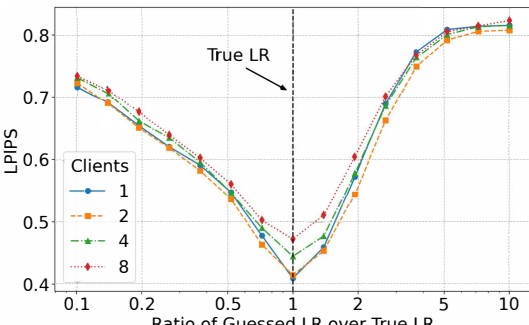

Figure 11: Quality of the raw reconstructed images when the attacker incorrectly guesses the global model's learning rate (LR). Image quality improves as the attacker guesses more correctly.

ever, within an order of magnitude of the true learning rate, the degradation follows a simple polynomial pattern. An attacker with sufficient computational resources can refine their guess over multiple iterations to improve reconstruction quality if the true learning rate is not known.

**Nonuniform Learning Rate.** The proposed attack relies on each client's gradient update being scaled by the same learning rate and this is also necessary for the global model to converge with FedAvg. To achieve the best model convergence during federated training, FedAvg scales each client's update by the client's number of training images, which gives the individual gradient of each training image equal weight in the global model update. If clients used very different learning rates, the clients with larger learning rates would dominate the global weight update, leading to suboptimal convergence. This is the basis for the assumption that the clients' learning rate in each round is either set by the server or otherwise controlled. For example, the clients may use a learning rate scheduler but agree

on its parameters so the scale of their updates does not vary significantly in a given round. Regardless of how the learning rate is set during federated training, scaling individual image gradients unevenly in a way that would disrupt the attack is also likely to impede the global model's convergence.

**Unknown Number of Images.** The proposed attack assumes that the attacker can correctly guess the total number of training images $N$ in a given round. This assumption simplifies the attacking algorithm but is not always needed. Instead, the attacker can search for this integer value and decide on the best guess leading to a successful recovery of the attacker's own training images. If both the learning rate and number of images are unknown, a joint parameter search can be conducted.

**Application to Secure Aggregation.** We evaluate the similarity between our approach and server-side attacks against the secure aggregation protocol, identifying both significant differences and an additional application scenario of our attack. Our problem of inverting the aggregated gradients of multiple clients is similar to the problem server-side attackers encounter in systems using the secure aggregation protocol, which prevents the parameter server from knowing individual clients' gradients (Bonawitz et al., 2017). Despite this similarity, we have not found any other works that obtain high-quality reconstructions without modifying the global model (Shi et al., 2023; Zhao et al., 2024) or relying on additional information the server might have about the client devices, such as device type and available memory (Lam et al., 2021), which would not be possible for a client attacker. Most of these attacks also rely on information collected across many training rounds, which may not be available to a client who cannot choose which rounds it is selected to participate in. In contrast, our attack does not require the attacker to disrupt the training protocol or have any information about the other clients beyond the model updates and total number of training images, which it may be able to guess. It also relies only on information from two consecutive training rounds. This indicates that our attack could also be performed by a server against a securely aggregated gradient and would allow it to avoid modifying the global model, maintaining the honest-but-curious threat model.

## C ASSUMPTIONS

Table 1: Many of the assumptions necessary for the proposed attack are shared by server-side gradient inversion attacks. We compare the assumptions necessary for our attack to ROG (Yue et al., 2023), DLG (Zhu et al., 2019), and iDLG (Zhao et al., 2020) to clarify which are unique to the curious-client threat model. Beyond what is required for these server-side attacks, the proposed attack requires that the number of clients in each training round is small and that the attacker knows or can guess the total number of images in a given round.

| Assumption | Ours (client) | ROG (server) | iDLG (server) | DLG (server) |
|---|---|---|---|---|
| Application: cross-silo/cross-device | cross-silo | both | both | both |
| Analytical label inversion | ✓ | ✓ | ✓ | |
| Single image per gradient | | | ✓ | |
| Each client trains on a single batch in each round | ✓ | ✓ | ✓ | ✓ |
| Clients can guess the total number of images in a given training round | ✓ | | | |
| Small number of clients | ✓ | | | |
| Small number of local iterations | ✓ | ✓ | ✓ | ✓ |
| Small number of images in each training round | ✓ | ✓ | ✓ | ✓ |
| Attacker has resources for complex attack | ✓ | ✓ | ✓ | ✓ |

## D    EFFECT OF UNEVEN LOCAL BATCH SIZE

We compare the performance of the proposed attack with uneven client batch sizes to confirm that the proposed attack is not affected when training examples are distributed unevenly between clients. To evaluate this, we distribute a total of 256 training images unevenly across clients, using an average client batch size of 16 images. Half of the clients are initialized with 21 images (two-thirds of the total training data), while the other half receive 11 images (one-third of the total). Figure 12 compares the image reconstruction quality between this uneven distribution and a system where each client has an equal batch size of 16 images, keeping the total number of training images constant. The evaluation is conducted with an even number of clients ranging from 2 to 8. The results indicate negligible differences in reconstruction quality between the two systems. This finding supports our hypothesis that the weighting behavior of FedAvg renders the attack robust to uneven batch size distributions.

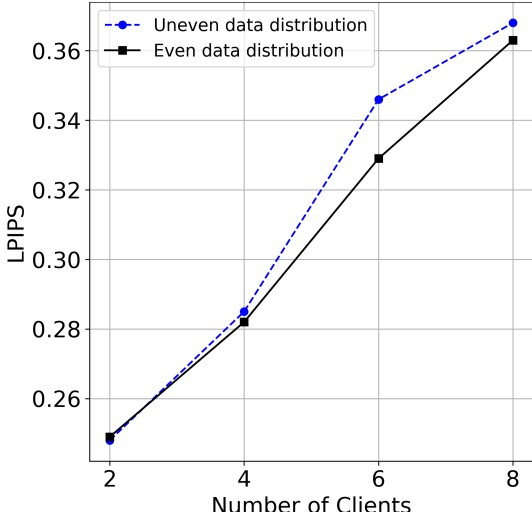

Figure 12: Quality of image reconstructions when training images are distributed unevenly between clients compared to an even distribution of training images. The effectiveness of the proposed attack is not sensitive to uneven client batch sizes, even when training for multiple local iterations.

## E    EFFECT OF INVERSION LEARNING RATE

Figure 13 examines the sensitivity of the proposed attack to variations in the attacker's inversion learning rate, which is used optimize the dummy data. We evaluate reconstruction quality by varying both the inversion learning rate and the number of clients where the FL learning rate, used to update the global model, is fixed at 0.03. The results show only minor differences in image reconstruction quality across different learning rates, with slight variations in the optimal learning rate depending on the number of clients. Overall, the attack's performance remains robust as long as the inversion learning rate is within a reasonable range.

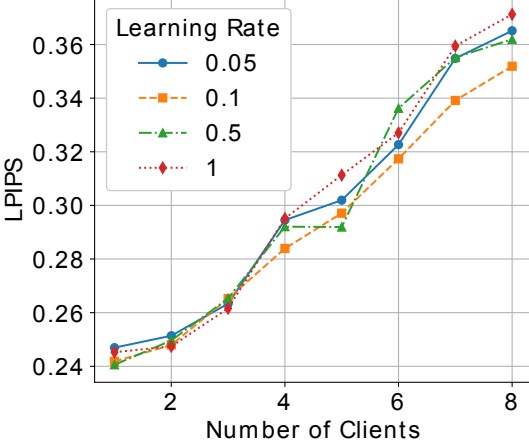

Figure 13: LPIPS of reconstructed images with varying attacker's inversion learning rate, which the reconstruction quality is not sensitive to.

## F    FURTHER ANALYSIS ON NUMBER OF LOCAL TRAINING ITERATIONS

We experimentally examine an upper bound of reconstruction performance for the proposed attack to better understand the effect of larger numbers of clients and local iterations. The attacker is initialized to simulate the true global model update by explicitly modeling $K$ clients. While this attack is not practical as it requires the attacker to correctly group the inverted image labels by client index $k$, it is useful for understanding how the number of local training iterations uniquely impacts the performance of client-side gradient inversion. Figure 14 presents the quality of the attacker's reconstructed images as a function of local iterations, with separate plots for simulations of 2, 4, 6, and 8 clients. The figure highlights the gap between the real attack (single-client simulation) and the upper bound, where the attacker correctly simulates the true number of clients. The results show that explicit multi-client simulation enables the attack to be unaffected by more local iterations, demonstrating that the discrepancy between the true update rule

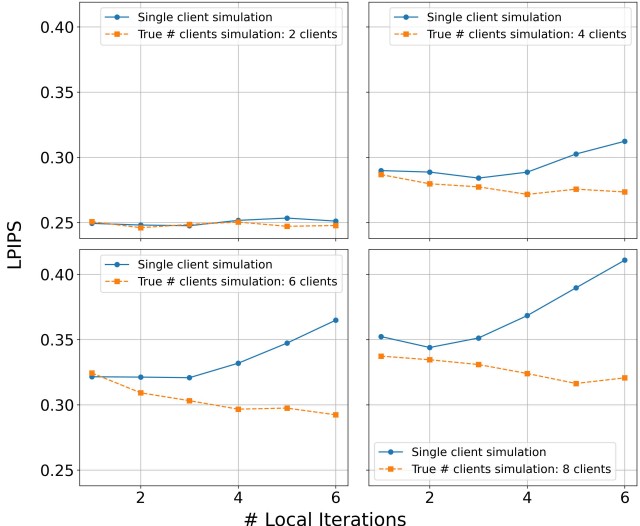

Figure 14: Reconstruction quality vs. the number of local iterations for simulations with 2 clients (upper left), 4 clients (upper right), 6 clients (lower left), and 8 clients (lower right). Each plot compares the proposed attack (single-client simulation) to a theoretical upper bound where the attacker explicitly simulates the true number of clients with labels grouped correctly. The gap between the proposed attack and the upper bound increases with more clients and local iterations, as the attacker's approximation error increases.

and the attacker's approximation accounts for the degradation in reconstruction quality as local iterations increase. Further, as the number of clients grows, the performance gap widens. Our result in Appendix A demonstrates that the attacker's single super client-based loss function becomes less accurate with more local iterations. This could be one explanation of why the reconstruction quality decreases as the local iteration count increases.

## G    ADDITIONAL POSTPROCESSOR COMPARISON TO ROG

Figure 15 compares the performance of our direct postprocessor to the GAN used by Yue et al. (2023) in both LPIPS and SSIM, as described in the experimental results section of the main text.

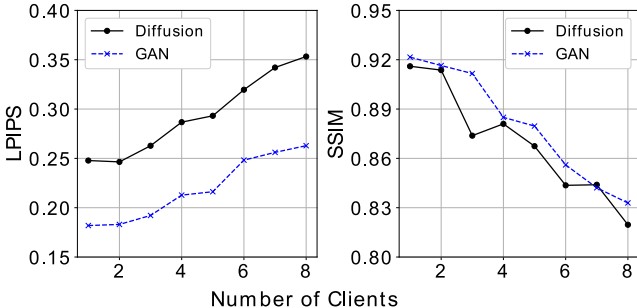

Figure 15: LPIPS and SSIM of reconstructed images using our direct postprocessor vs. GAN postprocessing with varying number of clients. Our method results in higher LPIPS as it blurs uncertain image details, compared to sharper but potentially less accurate outputs from GANs. SSIM, less sensitive to blurring, remains comparable across both approaches.

## H ADDITIONAL RECONSTRUCTION RESULTS

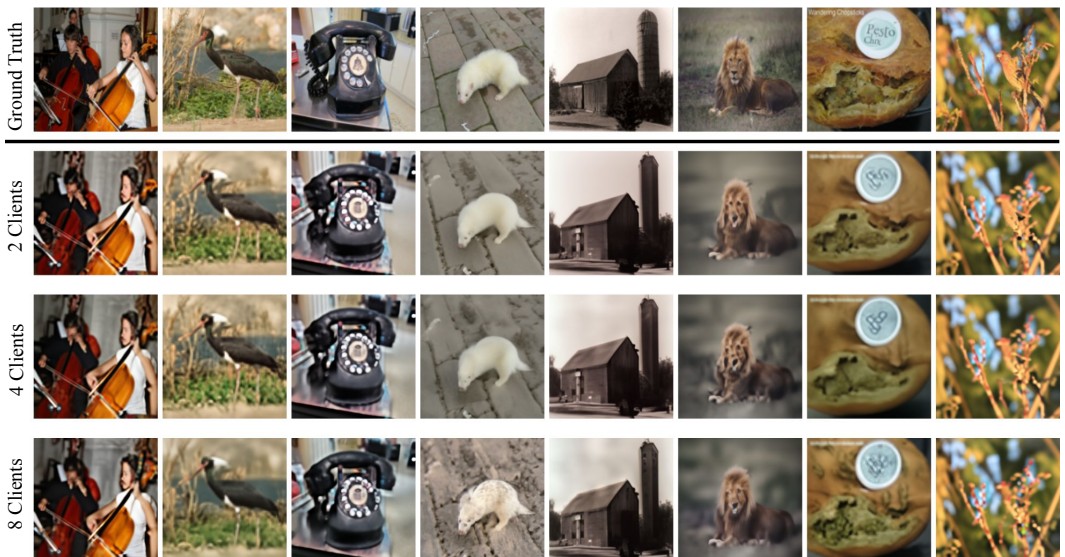

Figure 16: Images reconstructed by the proposed attack on a system using LeNet as the global model with 8 images per client and three local iterations.

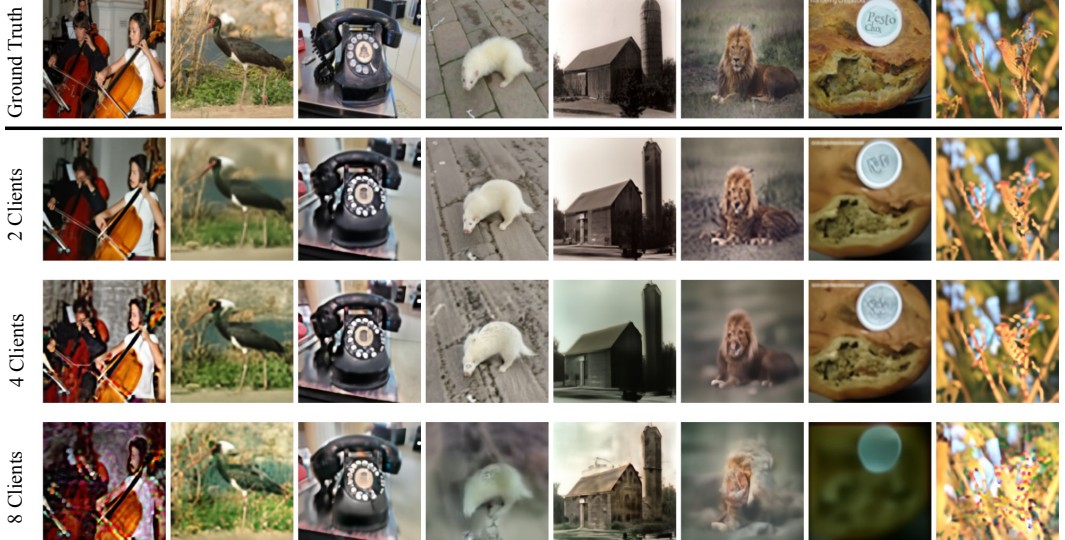

Figure 17: Images reconstructed by the proposed attack on a system using LeNet as the global model with 32 images per client and three local iterations.

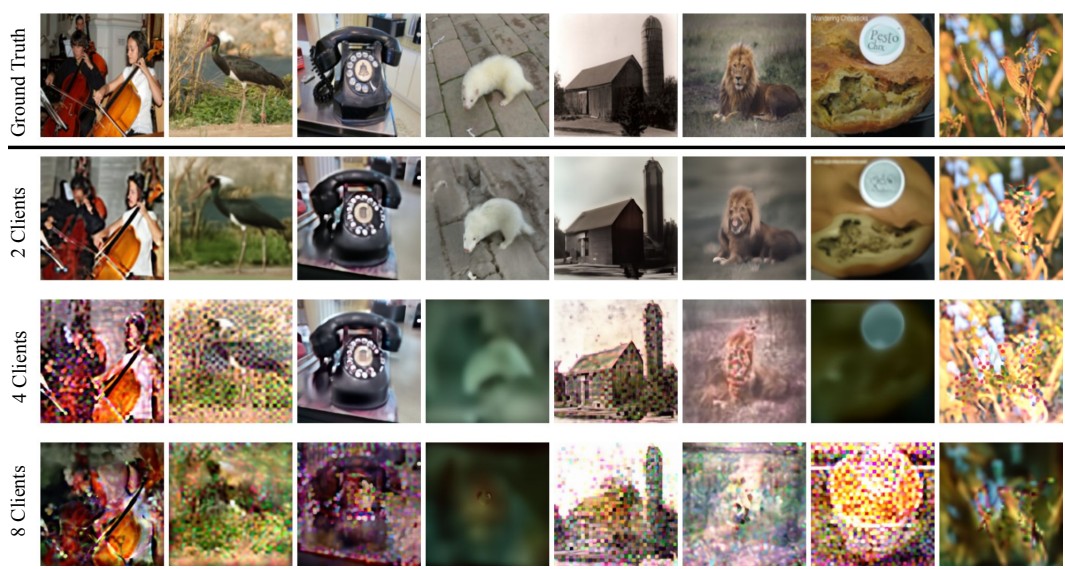

Figure 18: Images reconstructed by the proposed attack on a system using LeNet as the global model with 64 images per client and three local iterations.

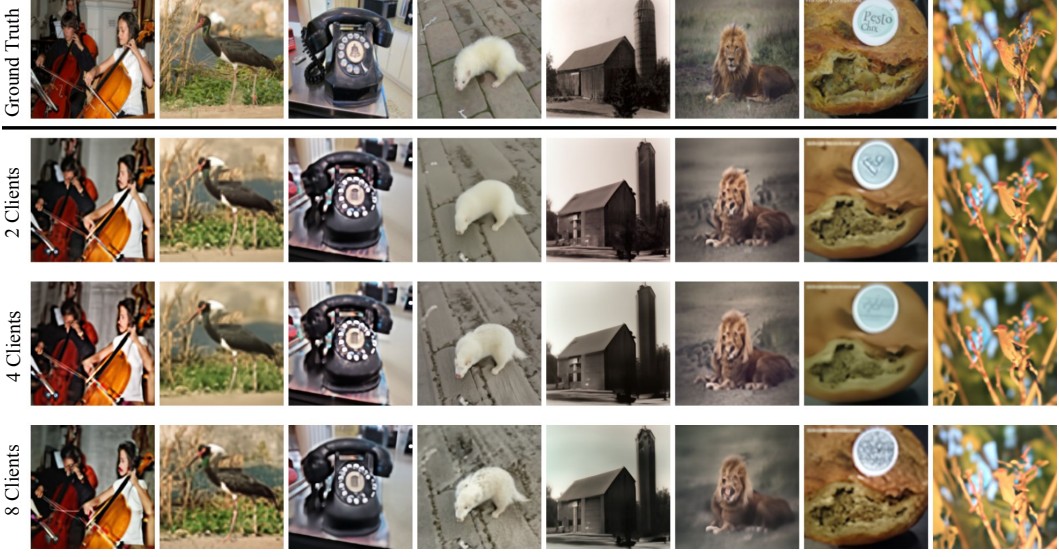

Figure 19: Images reconstructed by the proposed attack on a system using LeNet as the global model with 16 images per client and one local iteration.

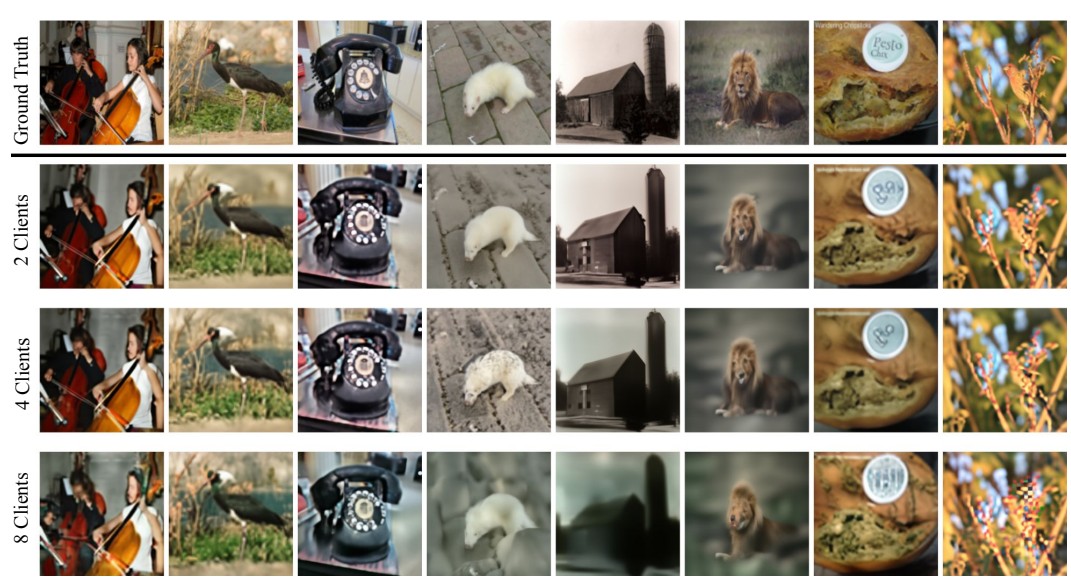

Figure 20: Images reconstructed by the proposed attack on a system using LeNet as the global model with 16 images per client and five local iterations.

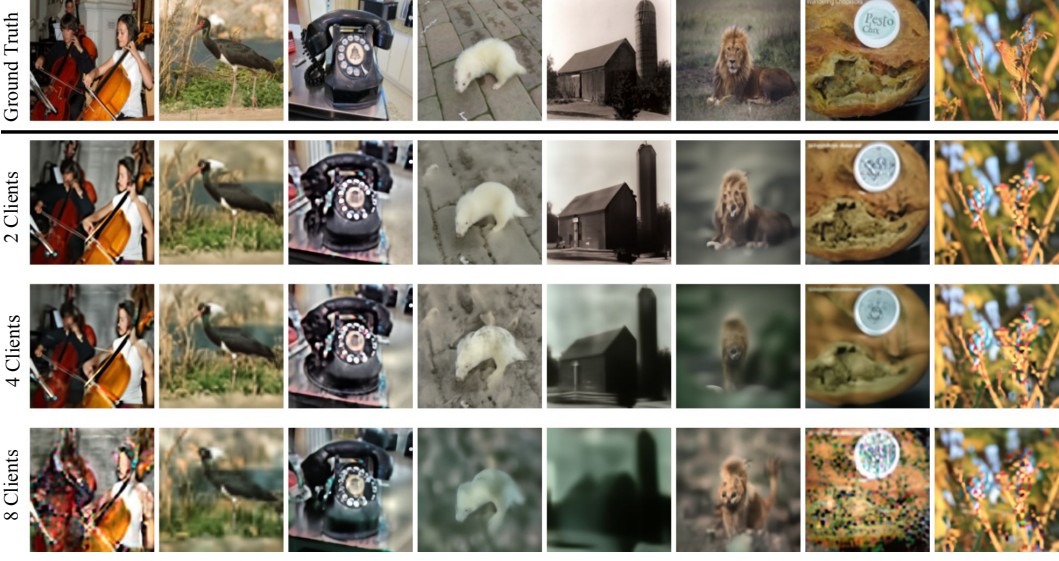

Figure 21: Images reconstructed by the proposed attack on a system using LeNet as the global model with 16 images per client and eight local iterations.

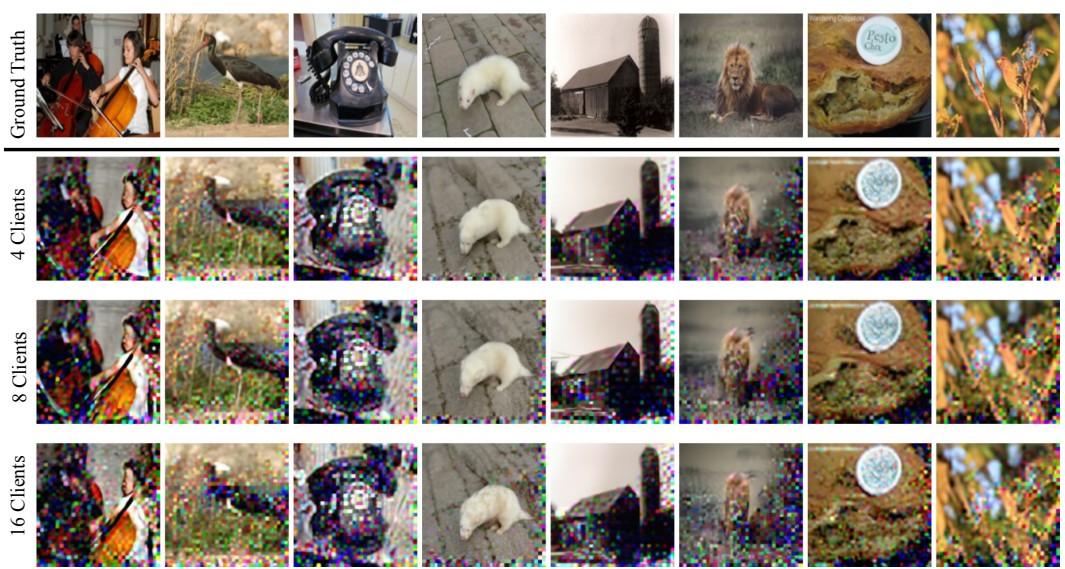

Figure 22: Images reconstructed by the proposed attack on a system using ResNet9 as the global model with 16 images per client and three local iterations.

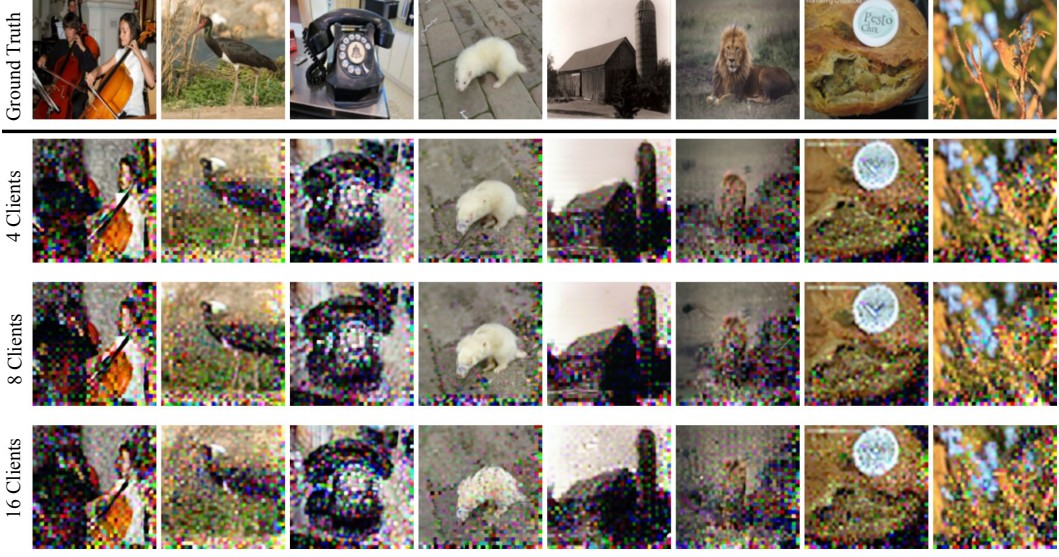

Figure 23: Images reconstructed by the proposed attack on a system using ResNet18 as the global model with 16 images per client and three local iterations.

