# OpenReview forum: "Federated Learning Nodes Can Reconstruct Peers’ Image Data"
_ICLR.cc/2026/Conference — ICLR 2026 Conference Withdrawn Submission_

### Official Review · Reviewer_ogD4 · 2025-10-22

**Soundness:** 2
**Presentation:** 3
**Contribution:** 2
**Rating:** 2
**Confidence:** 4

**Summary:**

The paper proposes an privacy attack on FL from the client side. Particularly, using the information on the gradient's update, the method solves an optimization problem for a rough estimation of the data. Then, they use diffusion models to further reconstruct the data.

**Strengths:**

The paper consider an important and interesting problem. The method is demonstrated to work in some specific settings.

**Weaknesses:**

Generally, given the results from the paper, I think it is challenging to do inversion attack from client side in FL. This is coming from the long list of assumptions such as knowing labels Y, knowing the number of total samples, low number of total samples , small models and  same training setting for all clients. Despite that, the paper chooses to focus on showing successful attacks on those limited setting. Furthermore, if any defense mechanism such as DP are applied, I do not think the attack will work. Thus, I would like to see a clear discussion and results on how each assumption would affect the feasibility of the attack.

Furthermore, another important results should be reported is the raw quality of the optimization in Eq.3, since it heavily dictates the performance of the post processing.

My understanding is that the method has 2 components, the optimization Eq.3 and the post-processing. While the optimization has appeared in many GI attack, the post-processing using diffusion models is also not novel as many research had been using that for denoising. I do not see the clear novelty of this work.

**Questions:**

see weaknesses

---

### Official Review · Reviewer_ZPPR · 2025-10-27

**Soundness:** 3
**Presentation:** 3
**Contribution:** 2
**Rating:** 2
**Confidence:** 4

**Summary:**

This paper demonstrates a privacy attack in Federated Learning (FL) where an "honest-but-curious client" can reconstruct the private training images of its peers.
This paper propose to use two diffusion models for gradient inversion and post-processing.

**Strengths:**

1. The authors analyzed the similarity and difference between server-side setting and client-side setting theoretically.

2. The method is strongly supported by thorough Empirical Study. The authors provide a comprehensive set of experiments to validate the attack's feasibility. They systematically analyze performance under various conditions, including an increasing number of clients, different batch sizes, and a growing number of local iterations . This empirical analysis clearly illustrates the conditions under which the attack succeeds or fails.

**Weaknesses:**

There are concerns about the significance of client-side setting, experiments, assumptions, and implications of lemma 1.
Please check questions below for details.

**Questions:**

1. Significance of the 'Client-Side' Setting is Unclear: The paper's framing makes it difficult to appreciate the unique importance of the client-side setting. By approximating the problem as a server-side attack on a large super-client batch, the problem looks like a large-batch server-side attack. Then, it is difficult to understand the significance of client-side setting.

2. Missing Baseline Comparison: The paper claims the client-side attack is more difficult but provides no direct comparison. A fair baseline would be to compare the proposed attack (e.g., $K=4$ clients, $N_k=16$ batch) against a standard server-side attack on an equivalent single batch (e.g., $N=64$). This is a key omission for validating the paper's claims.

3. Impractical Assumptions: The threat model relies on very strong assumptions. The attacker is assumed to correctly guess all corresponding distinct labels[A]. The assumption was justified because the batch size was small and the number of classes is large (1000 for imagenet). Howerver, this seems highly impractical in the tested scenarios, such as inverting an aggregated gradient from 512 images ($N=16, K=32$).

4. Unclear MDE Optimization: The paper states the "Best-quality Epoch" for the MDE postprocessor is found via visual inspection. It is unclear how an attacker, lacking the ground-truth images, could reliably select this optimal point before the image quality degrades.

5. Significance of Lemma 1: The finding that reconstruction degrades with more local iterations ($\tau$) (Figure 6)  is a known phenomenon[B]. The paper should more clearly articulate the novel contribution and implication of Lemma 1.


[A] Bo Zhao, Konda Reddy Mopuri, and Hakan Bilen. iDLG: Improved deep leakage from gradients.
arXiv preprint arXiv:2001.02610, 2020.

[B] Jonas Geiping, Hartmut Bauermeister, Hannah Dröge, and Michael Moeller. Inverting gradients-how easy is it to break privacy in federated learning? Advances in Neural Information Processing Systems, 33:16937–16947, 2020.

**Details Of Ethics Concerns:**

The paper is about attack in federated learning setting.

---

### Official Review · Reviewer_vLkB · 2025-10-31

**Soundness:** 2
**Presentation:** 3
**Contribution:** 2
**Rating:** 4
**Confidence:** 4

**Summary:**

This paper presents a client-to-client data stealing attack in Federated Learning (FL) based on gradient inversion. The authors theoretically prove the convergence of an algorithm that enables a client to infer another client’s update from two consecutive global model weights. Leveraging this insight, a malicious client can reconstruct other clients’ training images without interfering with the training process or accessing private system settings, such as the total number of clients or individual client information. A diffusion-based image restoration method is further introduced to enhance the reconstruction quality.

**Strengths:**

1.Realistic Threat Model: The paper considers a client-side attack scenario, which is arguably more realistic and harder to defend against than server-side attacks, making this threat setting highly relevant to real-world FL applications.
2.Rigorous Theoretical Foundation: The authors provide a complete convergence proof for the proposed gradient-inverse algorithm, demonstrating that a client can theoretically recover the true gradient from successive global model updates.
3.Refined image quality: The proposed MDE module leverages a pre-trained diffusion model to refine the reconstructed images, leading to improved visual fidelity compared to standard gradient-inversion methods.

**Weaknesses:**

W1.Lack of sufficient comparison with critical baseline: FedInverse [1] already demonstrated client-to-client data stealing prior to this work, making it a critical baseline. However, the paper only includes a limited qualitative comparison on MNIST (Fig. 9), which does not adequately establish the advantages of the proposed method over this prior approach.
W2.Lack of evaluation under heterogeneous client data distributions: Heterogeneous (non-IID) data partitions are widely studied and fundamental in FL, the paper does not investigate whether the proposed attack remains effective when client data distributions differ, limiting the understanding of the method’s applicability in realistic FL scenarios.
W3.Reliance on potentially unreliable similarity metrics: In Fig. 8, the paper claims its post-processing yields “more recognizable and more accurate image details” than ROG, yet simultaneously reports worse LPIPS/SSIM in Fig. 15 “due to blurring.” This inconsistency suggests LPIPS/SSIM alone may not reliably measure data-stealing success, and the authors should instead adopt metrics tailored to privacy leakage or copy detection (e.g., SSCD [2] or an attack-success score proposed by [3]) to more accurately assess whether reconstructed images constitute true data extraction.
W4.Lack of evidence that the FL training setup is realistic and meaningful: The paper does not demonstrate that the FL configuration used for evaluating the attack corresponds to a reasonable or practically relevant training setting. To justify that the FL model is being trained in a meaningful regime rather than simply overfitting, the authors should report the global model’s classification accuracy under their setup.
W5.Lack of effectiveness evaluation on privacy-sensitive datasets: While the paper argues for stronger privacy-preserving mechanisms in FL, it does not examine whether the attack remains effective under data distributions characteristic of privacy-critical domains (e.g., facial datasets like CelebA [4] or medical imaging such as COVID-19 CT scans [5]). Since these domains exhibit different distribution structures and visual statistics from ImageNet/MNIST, omitting such evaluations limits the understanding of the method's robustness and applicability across real-world FL scenarios.
[1] FedInverse: Evaluating privacy leakage in federated learning, ICLR’2024.
[2] A Self-Supervised Descriptor for Image Copy Detection, CVPR’2022
[3] Extracting Training Data from Diffusion Models, UsenixSecurity’2023
[4] Large-scale CelebFaces Attributes (CelebA) Dataset, ICCV’2015
[5] https://www.cancerimagingarchive.net/collection/ct-images-in-covid-19

**Questions:**

Q1.Could the authors clarify what visual quality corresponds to different LPIPS ranges—for example by providing representative reconstructed images for LPIPS bins (e.g., <0.05, 0.05–0.10, 0.10–0.20, >0.20)—and state/justify an LPIPS threshold above which a reconstruction should not be considered a successful data-stealing instance?
Q2. How sensitive is the optimization to the choice of the initial noise vector $X_T$ in Eq. 4? Specifically, how does the initialization affect (a) the number of epochs required to reach the best-quality reconstruction and (b) the final image fidelity?

---

### Official Review · Reviewer_G1vN · 2025-11-01

**Soundness:** 3
**Presentation:** 3
**Contribution:** 3
**Rating:** 4
**Confidence:** 3

**Summary:**

This paper investigates privacy leakage in cross-silo federated learning (FL).
Unlike prior work that assumes a malicious server, the authors demonstrate that even a client can reconstruct private training images of other clients by observing consecutive global models. The client computes the difference to estimate the averaged global gradient (the “super-client approximation”) and performs gradient inversion to recover visual data. To enhance quality, the authors apply two diffusion-based postprocessors: (1) a direct DDRM-based denoiser, and (2) a novel Masked Diffusion Enhancer (MDE), which optimizes the diffusion model’s latent noise to produce photorealistic reconstructions.

Experiments demonstrate that meaningful images can be recovered under realistic cross-silo FL settings.

**Strengths:**

+ This work shifts attention from server-side to client-side attackers, which is an underexplored but practically relevant risk — especially in systems that already deploy secure aggregation or encrypted communications.

+ The “super-client approximation” is conceptually simple and effective for recovering the aggregated gradient.

+ Extensive experiments convincingly show that privacy leakage remains significant even when the server is honest.

+ The diffusion-based postprocessing substantially improves perceptual fidelity; the MDE component yields visually impressive results.

+ Clear presentation

**Weaknesses:**

- Algorithmic novelty is modest. The core mechanism (computing global weight differences and running gradient inversion) is identical to what a server could already do. The main innovation is therefore the attacker identity, not the attack technique itself.

- The applicability is limited. It works mainly in cross-silo FL (few clients, small τ). In cross-device FL with hundreds of participants, the averaged gradients can dilute individual information and reconstructions degrade severely.

- Evaluation scope is narrow. Only image datasets and CNNs; no exploration of text or non-visual modalities where the same logic might (or might not) apply.

- More exhaustive experiments and real-world setups of FL should be explored as done in the following paper (inspired  by Office 31):

"Federated Learning for Commercial Image Sources", WACV 2023.

Dataset link: https://drive.google.com/file/d/1qgpj1TsGT4lnhhOmwR4gqVRigoHnMRnX

**Questions:**

Can you consider having a larger number of clients (say 32 or 64) to demonstrate the attack’s practical limits?

---

### Note · Authors · 2025-11-16

I have read and agree with the venue's withdrawal policy on behalf of myself and my co-authors.